# SEQPATE: DIFFERENTIALLY PRIVATE TEXT GENERATION VIA KNOWLEDGE DISTILLATION

## ABSTRACT

Protecting the privacy of user data is crucial when training neural text generation models, which may leak sensitive user information during generation. Differentially private (DP) learning algorithms provide guarantees on identifying the existence of a training sample from model outputs. PATE is a DP learning algorithm that fits large models well, such as GPT. In this paper, we propose SeqPATE that adapts PATE to text generation while satisfying DP. There are two key challenges in adapting PATE to text generation: (i) obtaining sequence-level supervision for text generation, and (ii) reducing noise required to protect the privacy given a large output space (i.e. vocabulary size). For (i), we generate pseudo input and reduce the sequence generation problem to the next word prediction. For (ii), we reduce the output space with top-$k$ and top-$p$ selection strategy that dynamically filters the candidate words; and we refine the teacher aggregation of PATE to avoid the low agreement rates due to voting over the large output space. To limit the privacy loss, we design an efficient knowledge distillation to reduce the frequency of distilling from the private data. We apply SeqPATE to a simple text generation task (sentence completion) and achieve 39% and 28% gains in Bleu4 on two datasets.

## 1 INTRODUCTION

Recent work showed that sensitive user information in training corpora, such as address and name, can be extracted from text generation models (Carlini et al., 2019; 2020). Providing privacy guarantees to the text used to train text generation models has become a critical problem. Differential privacy (DP) provides provable guarantees against the identification of individuals in the dataset. Deep learning models with DP guarantees ensure the existence of a specific training sample cannot be detected.

The most popular DP algorithm for deep learning is Noisy-SGD (Song et al., 2013; Bassily et al., 2014; Abadi et al., 2016), which adds noise to the gradients. However, the advantage of NoisySGD is weaken as the model becomes larger (Yu et al., 2020), and NoisySGD requires a per-example gradient clip, which causes non-convergence and system overheads (Zhu et al., 2020; Bu et al., 2021). Applying Noisy-SGD to large text generation models (e.g. GPT-2) requires additional tricks such as memory reduction (Li et al., 2021) and advanced fine-tuning strategies (Yu et al., 2021). PATE (Papernot et al., 2017) is a DP learning model using a teacher-student framework: a student accessing non-sensitive data distills knowledge from the aggregated predictions of multiple teachers trained on sensitive data. Calibrated noise is added to the aggregated predictions to satisfy DP. PATE can handle large models since its privacy cost derives from the knowledge distillation instead of the model parameters. PATE takes advantage of non-sensitive data which is also available in our scenario.

In this paper, we propose, SeqPATE, a DP learning algorithm for text generation that exploits both sensitive and non-sensitive data by knowledge distillation. Specifically, we adapt PATE for text generation. The adaptation is challenging due to the sequential generation and the large output space (i.e. vocabulary size) in text generation. Firstly, to obtain the sentence-level supervision for text generation, directly applying PATE needs to roll out all teachers to produce a sentence (i.e. all teachers must vote to generate a word, which is then used as the input for the next word prediction). Running inference for a large number of teachers can be costly. Secondly, the large output space results in (i) low agreement rate among teachers and (ii) large noise required by DP, both would significantly hurt task performance.

To avoid sequential generation, we generate pseudo data using a pre-trained language model such that teachers only need to provide token-level supervision given the pseudo input. To handle the large output space, we aggregate teachers' outputs by interpolating their output distributions instead of voting with argmax predictions. To reduce the noise scale, we propose strategies to dynamically filter candidate words that only keep words with large probabilities. Further, we design an efficient knowledge distillation strategy that only queries the teacher when the student has poor performance.

We evaluate our method on sentence completion, a simple but representative task of text generation. Compared to baselines including Noisy-SGD (Kerrigan et al., 2020), SeqPATE improves Bleu4 by 39% and 28% on two datasets with meaningful privacy protection [1]. We observe SeqPATE works particularly well under relatively strong privacy protection (e.g., $\varepsilon \leq 2$) compared to Noisy-SGD. Our contribution is threefold: (i) Our model significantly surpasses baselines and makes DP algorithm achieve satisfactory performance on text generation; (ii) We propose several practical strategies to enable SeqPATE to handle text generation with a sequential of classifications over larger output spaces; (iii) We are the first to adapt PATE, an effective DP algorithm, to text generation.

## 2 Problem Setup

Our goal is to achieve the privacy protection defined by DP in text generation: preventing attackers from inferring whether a sample or an n-gram appears in the training set. (Formal definition in Sec. 5)

PATE assumes students are trained on unlabeled non-sensitive data. Similar to PATE, our setting contains two textual datasets: 1. a private set $\mathcal{D}^{\text{pri}}$ from a corpus with sensitive information. 2. a public set $\mathcal{D}^{\text{pub}}$ without sensitive information (e.g. incomplete or masked text spans) or data contributors (e.g. volunteers) have no objection to publishing their data. Algorithms are required to protect the privacy on the private set and can ignore the privacy protection on the public set. The final goal is to do well on the private set during inference.

Text generation is to generate a sequence of words $S = \{w_1, w_2, \ldots, w_l\}$. Our application is a simple text generation task, sentence completion, which aims to generate the remaining part of a sentence $w_{1:l}$ given its prefix $p_{1:l'}$, where $p_{1:l'}$ and $w_{1:l}$ are the abbreviation of $\{p_1, p_2, \ldots, p_{l'}\}$ and $\{w_1, w_2, \ldots, w_l\}$ respectively. $p_{1:l'}$ and $w_{1:l}$ compose a complete sentence. We collect some input prefixes $p_{1:l'}$ to compose the public set $\mathcal{D}^{\text{pub}}$ and the original samples (complete sentences) compose the private set $\mathcal{D}^{\text{pri}}$. Such a setting fits for some real-world text generation applications: in dialog systems, the training samples from online services consist of questions and responses. The questions from customer service staffs or service robots can be public and the response from users carrying individual information should be private. In writing assistants where a sample has an input prompt and an essay body, the prompt can be public but the body written by data contributors is private.

## 3 Background on DP and PATE

Differential privacy (DP) (Dwork et al., 2006; 2014) is a quantifiable definition of privacy that provides provable guarantees on identifications of individuals in the dataset. ML algorithms with DP guarantee ensure that each individual training sample has a degree of *plausible deniability*, i.e., the trained model is *just as likely as* to have been trained on an alternative dataset *without* that sample.

PATE, designed for classification tasks, takes advantage of an unlabeled public dataset $\mathcal{D}^{\text{pub}}$ and also trains on a labeled private $\mathcal{D}^{\text{pri}}$ in a semi-supervised scenario. PATE is model-agnostic and treats models as black-boxes. PATE (Papernot et al., 2017) achieves DP via a teacher-student framework with $M$ teacher models and a student model, where the student learns from the private data via knowledge distillation through teachers. PATE consists of three parts,

- **Teacher models** are trained on the private set $\mathcal{D}^{\text{pri}}$. $\mathcal{D}^{\text{pri}}$ is shuffled and divided into $M$ disjoint subsets, and $m$-th subset serves as the training set for $m$-th teacher model $f_\phi^m$.

- **Teacher aggregation** merges teachers' outputs. After teachers' training, each teacher $f_\phi^m$ conducts the prediction (classification) on student's training set $\mathcal{D}^{\text{pub}}$. To satisfy DP, PATE aggregates $M$

---

[1] We choose $\varepsilon = 2$. DP with $\varepsilon \in [0.1, 5]$ is regarded as a meaningful protection (Triastcyn & Faltings, 2020).

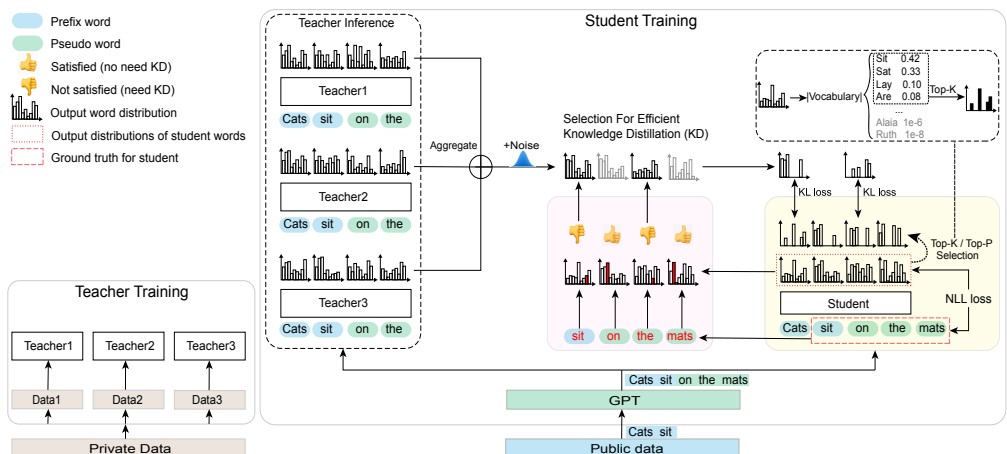

Figure 1: Overview of SeqPATE. SeqPATE trains teachers on the private data; it conducts student training and teacher inference on pseudo sentences generated by GPT based on the public data. The student is supervised by the aggregation of teacher output distributions and benefits from efficient knowledge distillation (pink block) and top-$k$ / top-$p$ selection (white block in the top right corner).

teachers' by voting with their top-1 predictions. PATE adds noise to aggregated teacher outputs so that attackers cannot access the exact outputs to infer original information in samples.

- **A student model** $f_\theta$ is trained on the public set $\mathcal{D}^{\text{pub}}$ and distills knowledge from private set (via the teachers) without accessing the private samples. In this way, the sensitive information in the private set is preserved even if the student's architecture and parameters are public or reverse-engineered by an adversary. We publish only the student model for prediction.

## 4 APPROACH

Fig. 1 shows an overview of the SeqPATE. Given the public prefix (e.g., "Cats sit"), we first obtain pseudo inputs by generating the remaining part of the whole sentence (e.g., "Cats sit on the mats") using a pre-trained language model (Sec. 4.1). The teachers' output distributions are then averaged to provide word-level supervision on the whole sentence (Sec. 4.2). To reduce the noise over large output space and limit the privacy loss, we propose top-$k$ / top-$p$ selection to dynamically filter the unimportant candidates and propose efficient knowledge distillation to reduce the number of teacher queries (Sec. 4.3). Our training algorithm is shown in Appendix A.

### 4.1 STUDENT MODEL

Conventional text generation models generate words in a sentence step-by-step and left-to-right. Naively applying PATE to the above paradigm requires all the teachers to be roll out to obtain supervision at every step, where all the teachers aggregate to generate a word at each step and the word acts as the next input for all teachers. This means we should conduct all teachers' inferences simultaneously and interdependently, which is costly in both computation (involving hundreds of teachers) and privacy costs.

Hence, in SeqPATE, we employ GPT to generate pseudo sentences and conduct teacher inference and student training on the pseudo sentences. Firstly, given a prefix in $\mathcal{D}^{\text{pub}}$, we employ the public released pre-trained GPT to conduct the inference to complete a pseudo sentence. All the generated pseudo sentences compose the student's training set, denoted as $\tilde{\mathcal{D}}^{\text{pub}}$. Secondly, we conduct teacher inference and student training on the sample $S = \{w_1, w_2, \ldots, w_{|S|}\}$ from $\tilde{\mathcal{D}}^{\text{pub}}$. At the $i$-th step, for the training of conventional text generations, the student learns to generate a word $\hat{w}_i$ given its previous words $w_{1:i-1}$. At the $i$-th step, our teachers conduct the inference to generate $\hat{w}_i$ given the input $w_{1:i-1}$ from the sample $S$. Then, each teacher provides its output for aggregation. All teachers share the same input with the student so that the outputs of teachers and student are aligned and comparable.

## 4.2 TEACHER AGGREGATION

The original PATE aggregates teacher results by taking the majority vote of the predicted labels from all the teachers. However, this aggregation strategy does not fit for the large output space in text generation. This is because the number of votes for each candidate may be very low and it would result in low agreement. For example, multiple candidates may tie for the top-1 prediction.

Therefore, inspired by (Hinton et al., 2015; Chen et al., 2020), we aggregate $M$ teachers' results by averaging their output distributions, where each teacher model is trained on one of the disjoint subsets divided from the private set $\mathcal{D}^{\mathrm{pri}}$. At the $i$-th step on teacher inference, the $m$-th teacher model $f_\phi^m$ predicts a word $\hat{w}_i$ with a probability $p_\phi^m(\hat{w}_i)$. Then, our model aggregates all teachers' probabilities by averaging $p_\phi(\hat{w}_i) = \frac{1}{M} \sum_{m=1}^{M} p_\phi^m(\hat{w}_i)$. Like most DP methods, our model adds noises so that the privacy impact can be analyzed and bounded. Different from PATE that adds noise to vote numbers, we add noise to the aggregation of teachers' output distributions. Following (Papernot et al., 2018), we apply the Gaussian mechanism which adds i.i.d. $\mathcal{N}(0, \sigma^2)$ to each coordinate, where $\sigma$ controls the strength of privacy protection (more details in Sec. 5).

## 4.3 SUPERVISION ON THE STUDENT MODEL

**Reducing output space via top-$k$ or top-$p$ selection.** Recall that the teacher output is a distribution over all words in the vocabulary, and the noise must be added to each coordinate to satisfy DP. To reduce the output dimension (hence the amount of noise), we only assign the probability mass to a subset of candidates $W_i$. For all teachers and the student, we assign zero probability mass to any candidates not in $W_i$ and re-normalize distribution as shown in Eq. 1, where $g(\cdot)$ is the normalizing function. This operation means the words not in $W_i$ will never occur in student's training and teachers' supervision, so those words do not need noise to provide protection.

$$g(p(\hat{w}_i \mid w_{1:i-1})) = \begin{cases} \dfrac{p(\hat{w}_i \mid w_{1:i-1})}{\sum_{w \in W_i} p(w \mid w_{1:i-1})} & \text{if } \hat{w}_i \in W_i \\ 0 & \text{otherwise.} \end{cases} \tag{1}$$

We determine the subset $W_i$ by two selection strategies: top-$k$ and top-$p$. In the top-$k$ selection, at $i$-th step, we select the top-$k$ candidates according to the probabilities generated by the student model. Further, inspired by nucleus sampling (Holtzman et al., 2019), we propose a top-$p$ selection that chooses the minimum $k$ where the cumulative probability of the top-$k$ words is larger than a constant close to 1 (e.g. 0.95) such that the candidate set covers a large proportion of the distribution. We note that the candidate selection is based on the student's output on public or already released input, thus not affecting the privacy guarantee. This choice improves the privacy-utility tradeoff by adaptively allocating the available privacy budgets to release the information more relevant to the task.

**Efficient knowledge distillation.** Reducing the number of querying teachers results in better privacy protection. Therefore, we acquire teacher supervision only when the student has poor performance. Concretely, we measure student's performance on each token by comparing its output distribution with the ground truth token (from $\tilde{\mathcal{D}}^{\mathrm{pub}}$). Let $r_{w_i}$ be the rank of the ground truth token $w_i$ in the student's predicted distribution, (i.e. the number of tokens with higher or equal probability). The student model acquires the teachers' supervision only if $r_{w_i}$ is larger than a certain threshold $r_c$. We note that the selection of tokens relies only on the student and is independent of the teachers, thus the selection does not cause additional privacy loss.

**Training objectives.** Different from the semi-supervised setting in PATE, our student model is supervised by both teachers' aggregation and labels from the pseudo public set $\tilde{\mathcal{D}}^{\mathrm{pub}}$, since text generation corpora are naturally labeled, unlike in classification tasks. Even if the pseudo set $\tilde{\mathcal{D}}^{\mathrm{pub}}$ provides only pseudo samples, the labels from those samples are also useful given that the number of querying teachers is limited due to privacy concerns. The student's loss function consists of two parts as shown in Eq. 2,

- $\mathcal{L}_i^{\mathrm{KL}}$ shows teachers' supervisions. The student learns from teachers by minimizing KL divergence between the student output distribution $p_\theta$ and the merged teacher output distributions $p_\phi$.

- $\mathcal{L}_i^{\mathrm{NLL}}$ shows the supervisions from the pseudo public set $\tilde{\mathcal{D}}^{\mathrm{pub}}$ generated by GPT. As conventional language models, our student model is trained by negative log-likelihood (NLL) loss on $\tilde{\mathcal{D}}^{\mathrm{pub}}$. The words $w_i$ in the training sample is naturally the label for the student $f_\theta(w_{1:i-1})$ at the $i$-th step.

$$
\begin{aligned}
\mathcal{L}_i^{\mathrm{KL}} &= \mathrm{KL}\left(g(\frac{1}{M}(\sum_{m=1}^{M} p_\phi^m(\cdot \mid w_{1:i-1}) + \mathcal{N}(0, \sigma^2))) \,\|\, p_\theta(\cdot \mid w_{1:i-1})\right), \\
\mathcal{L}_i^{\mathrm{NLL}} &= -\log p_\theta(w_i \mid w_{1:i-1}), \qquad \mathcal{L} = \sum_{S \in \tilde{\mathcal{D}}^{pub}} \sum_{i=1}^{|S|} (\mathcal{L}_i^{\mathrm{NLL}} + \lambda \mathcal{L}_i^{\mathrm{KL}}).
\end{aligned}
\tag{2}
$$

where $\mathcal{L}_i$ is loss at $i$-th step, $|S|$ denotes the length of sentence $S$, and $\lambda$ balances the two terms. Meanwhile, the noise scale $\sigma$ is discussed in Sec. 5. $g(\cdot)$ is the re-normalization function used in Eq. 1 that re-normalizes the aggregated probabilities on the selected top-$k$ or top-$p$ words with noise [2]. According to the efficient knowledge distillation strategy, at each step, the loss $\mathcal{L}_i$ is the combination of $\mathcal{L}_i^{\mathrm{NLL}}$ and $\mathcal{L}_i^{\mathrm{KL}}$ if the student needs teachers' supervision; otherwise, $\mathcal{L}_i = \mathcal{L}_i^{\mathrm{NLL}}$.

## 5 PRIVACY ANALYSIS

### 5.1 PRELIMINARY OF DIFFERENTIAL PRIVACY

Let $\mathcal{D}, \mathcal{D}'$ denote two neighboring datasets which differ at only one individual.

**Definition 1** (Differential privacy). *For $\varepsilon > 0$ and $\delta \geq 0$, a randomized algorithm $\mathcal{M} : \mathcal{X}^n \to \mathcal{Y}$ is $(\varepsilon, \delta)$-differentially private if for any neighboring datasets $\mathcal{D} \sim \mathcal{D}'$ and any $S \subseteq \mathcal{Y}$,*

$$
\Pr[\mathcal{M}(\mathcal{D}) \in S] \leq e^\varepsilon \cdot \Pr[\mathcal{M}(\mathcal{D}') \in S] + \delta
$$

The definition ensures that it is information-theoretically impossible for an adversary to infer whether the input dataset is $\mathcal{D}$ or $\mathcal{D}'$ even with arbitrary side information. The definition is also *future proof* thanks to its closure to post-processing.

**Lemma 2** (Post-processing). *If $\mathcal{M}$ obeys $(\varepsilon, \delta)$-DP, then for any function $f$, $f \circ \mathcal{M}$ is also $(\varepsilon, \delta)$-DP.*

One notable property of DP is that it automatically protects the privacy of groups of multiple units.

**Lemma 3** (Group privacy). *An $(\varepsilon, \delta)$-DP mechanism $\mathcal{M}$ on individuals is $(k\varepsilon, \frac{e^{k\varepsilon}-1}{e^\varepsilon-1}\delta)$-DP on groups of size $k$ for all integers $k \geq 1$.*

**Lemma 4** (Analytical Gaussian mechanism (Balle & Wang, 2018)). *For a numeric query $f : \mathcal{X}^n \to \mathbb{R}^d$ over a dataset $\mathcal{D}$, the randomized algorithm that outputs $f(\mathcal{D}) + Z$ where $Z \sim \mathcal{N}(0, \sigma^2 I_d)$ satisfies $(\varepsilon, \delta(\varepsilon))$-DP for all $\varepsilon \geq 0$ and $\delta(\varepsilon) = \Phi(\frac{\Delta}{2\sigma} - \frac{\varepsilon\sigma}{\Delta}) - e^\varepsilon \Phi(-\frac{\Delta}{2\sigma} - \frac{\varepsilon\sigma}{\Delta})$. where $\Delta := \Delta_2^{(f)} = \max_{\mathcal{D} \sim \mathcal{D}'} \|f(\mathcal{D}) - f(\mathcal{D}')\|_2$ is the global L2 sensitivity of $f$ and $\Phi$ is the CDF function of $\mathcal{N}(0, 1)$.*

The above is the $(\varepsilon, \delta)$-DP of a single Gaussian mechanism, and the following lemma shows that we can use the same result for an adaptive composition of a sequence of Gaussian mechanisms.

**Lemma 5** (Composition of Gaussian mechanisms (Dong et al., 2019)). *The adaptive composition of a sequence of Gaussian mechanisms with a noise level $\sigma_1, \sigma_2, \ldots$ and global L2 sensitivity $\Delta_1, \Delta_2, \ldots$ satisfies $(\varepsilon, \delta(\varepsilon))$-DP for all $\varepsilon \geq 0$ and $\delta(\varepsilon) \leq \delta_{\mathcal{M}}(\varepsilon)$ where $\mathcal{M}$ is a Gaussian mechanism with noise multiplier $\sigma/\Delta = \left(\sum_i (\Delta_i/\sigma_i)^2\right)^{-1/2}$.*

Specifically, the adaptive composition of a $k$ identical Gaussian mechanism with noise multiplier $\sigma$ satisfies the same privacy guarantee of that of a single Gaussian mechanism with a noise multiplier $\sigma/\sqrt{k}$. By fixing $k$ and $\varepsilon$, we can calibrate the noise by choosing an appropriate $\sigma$ in our algorithm.

---

[2]Mathematically, the input of the $g(\cdot)$ may be negative and we re-normalize it to 0. Practically, we observed being negative is extremely rare since the $M$ is very big (2000) and the first term dominates the input of $g(\cdot)$

## 5.2 DIFFERENTIAL PRIVACY AT THE SAMPLE LEVEL

Recall that we partition the private dataset to $M$ disjoint subsets, and train a teacher model on each subset. Let vector $x_i \in \mathbb{R}^{|\mathcal{V}|}$ denote the probability distribution predicted by the $i$-th teacher model for a specific word, where $|\mathcal{V}|$ is the vocabulary size. The function $f(\mathcal{D}) := \sum_{i=1}^{M} x_i$ is the sum of the probability distributions predicted over all teachers. Since the dataset is disjoint and also has no repeating sample, changing one sample will only affect one teacher model. For neighboring datasets $\mathcal{D}, \mathcal{D}'$, let $j$ denote the index of different teacher models, i.e., $x_j \in \mathcal{D}$ and $x'_j \in \mathcal{D}'$ are different. Then, the sensitivity $\Delta$ in Lemma 4 & 5 is (Detailed deduction in Appendix B),

$$\Delta := \Delta_2^{(f)} = \|f(\mathcal{D}) - f(\mathcal{D}')\|_2 \leq \|x_j - x'_j\|_2 \leq \sqrt{2},$$

Then, adding noise calibrated in Lemma 5 to each coordinate is sufficient to preserve $(\varepsilon, \delta(\varepsilon))$-DP for $f(\mathcal{D})$. Finally, when we extract top-$k$ coordinates of $f(\mathcal{D})$ (Sec. 4.3), the privacy guarantee also holds according to the post-processing property. Intuitively, the information about whether a sample occurs in the private training set is protected and the protection satisfies $(\varepsilon, \delta(\varepsilon))$-DP.

## 5.3 DIFFERENTIAL PRIVACY OF USER'S SECRET PHRASES

The above analysis focuses on DP at the sample level, but the privacy guarantee also applies to individual users (who contributed many samples), which prevents each user's *secret phrases* (or text spans), such as SSN number, phone number, and addresses from being memorized (or generated *as is*) by the model. Consider a secret phrase $t$ that occurs $s_t$ times ($s_t \geq 1$) in the private set. We obtain DP guarantee for each $t$ separately [3] in the following two scenarios. In the first scenario where the teachers' data is not partitioned by users, the protection on $t$ satisfies $(s_t\varepsilon, \frac{e^{s_t\varepsilon}-1}{e^\varepsilon-1}\delta)$-DP according to the group privacy lemma.

SeqPATE allows us to partition the teachers' training data by users. In this scenario, adding or removing a user (and all her samples) only affects one teacher, thus the same privacy guarantee we have derived for sample-level DP applies to the user level, too. As a result, our approach enjoys stronger guarantee than naively applying the group privacy lemma. Our approach depends only on the total number of teachers affected, which is bounded by $\tilde{s}_t := \min\{s_t, \# \text{ of users who know } t\}$. $\tilde{s}_t$ is often 2 or 3 (e.g., husband and wife, parents and child), even if $s_t$ is large. Clearly, adding or removing a phrase $t$ will result in a sensitivity $\sqrt{2}\tilde{s}_t$, and the exact $(\varepsilon, \delta(\varepsilon, \tilde{s}_t))$-DP for phrase $t$ can be obtained according to Lemma 4 & 5, where $\delta(\varepsilon, \tilde{s}_t) = \Phi(\frac{\tilde{s}_t}{\sqrt{2}\sigma} - \frac{\varepsilon\sigma}{\sqrt{2}\tilde{s}_t}) - e^\varepsilon\Phi(-\frac{\tilde{s}_t}{\sqrt{2}\sigma} - \frac{\varepsilon\sigma}{\sqrt{2}\tilde{s}_t})$. The above user-level protection is also one merit of SeqPATE that NoisySGD does not have.

**How does DP prevent memorization?** We remark that the protection against unintended memorization of a model follows from the definition of DP. Consider the attack by (Carlini et al., 2019), which uses a language model to predict a secret phrase $t$ given a prefix. By the closure to post-processing, the prediction also satisfies DP. Take $\mathcal{S}$ to be the undesirable event that the model correctly generates the phrase $t$. The definition of DP implies that the probability of $\mathcal{S}$ to happen when $t$ is part of the training data is at most $e^\varepsilon$ larger than the probability of an alternative model trained without $t$ in the data. The chances for the latter model to generate a sequence with $t$ is astronomically small, thus DP implies that the probability of $\mathcal{S}$ under the former model needs to be just as small.

# 6 EXPERIMENTS

## 6.1 EXPERIMENTAL SETTING

We evaluate our model on two datasets. AirDialog dataset (Wei et al., 2018) consists of 1M utterances from custom service dialog about flight booking; Europarl_v6 dataset [4] consists of 2M English sentences collected from European Parliament. Pre-trained GPT-2 (Radford et al., 2018) provides initial parameters for comparing methods. Before student training, we conduct additional fine-tuning on the public data $\tilde{\mathcal{D}}^{\text{pub}}$ via NLL loss to provide initialization. We evaluate the quality of generated

---

[3]A formal definition of this is called personalized differential privacy, first seen in (Ghosh & Roth, 2011)
[4]www.statmt.org/europarl

texts with *PPL* (Perplexity, the lower the better) and *B-n* (Bleu-n, the higher the better) (Papineni et al., 2002) and evaluate the performance of privacy protection with *P-n*, which is the fraction of unique n-grams in model generated texts that can be found in the private text (lower score indicates fewer privacy leaks). Our base model is GPT2-small and the default number of teacher models is 2,000. The batch size is 32 for all comparing methods and the $\lambda$ in Eq. 2 is 20. (See details about datasets, settings, metrics, implementation, and more experimental results in Appendix C to L).

## 6.2 OVERALL PERFORMANCE

| | | AirDialog | | | | | | Europarl_v6 | | | | | |
| | | PPL | B-1 | B-2 | B-3 | B-4 | P-3 | P-4 | PPL | B-1 | B-2 | B-3 | B-4 | P-3 | P-4 |
|---|---|---|---|---|---|---|---|---|---|---|---|---|---|---|---|
| No Protect | Pri-GPT | 3.88 | 42.49 | 28.26 | 21.51 | 17.16 | 0.94 | 0.91 | 23.25 | 12.97 | 4.55 | 1.77 | 0.86 | 0.85 | 0.65 |
| Public | Pub-GPT | 63.16 | 5.85 | 1.18 | 0.31 | 0.10 | 0.09 | 0.03 | 57.40 | 12.42 | 3.61 | 1.02 | 0.35 | 0.51 | 0.29 |
| Data Only | Pub-GPT+$\mathcal{D}^{pub}$ | 19.39 | 7.58 | 2.11 | 0.71 | 0.25 | 0.14 | 0.06 | 45.40 | 13.28 | 4.31 | 1.38 | 0.52 | 0.62 | 0.38 |
| DP | Pub-GPT+DP-SGD | 20.39 | 7.30 | 2.36 | 0.95 | 0.42 | **0.32** | 0.18 | 40.69 | 12.37 | 4.04 | 1.22 | 0.43 | 0.70 | 0.47 |
| Algorithms | Pub-GPT+DP-SGD+$\tilde{\mathcal{D}}^{pub}$ | 17.65 | 7.90 | 2.69 | 1.14 | 0.56 | 0.33 | 0.19 | 40.46 | 12.26 | 3.99 | 1.20 | 0.42 | **0.69** | 0.46 |
| ($\varepsilon = 2$) | SeqPATE (Ours) | **13.67** | **11.56** | **4.35** | **1.82** | **0.78** | **0.32** | **0.17** | **37.80** | **13.64** | **4.40** | **1.43** | **0.55** | **0.69** | **0.45** |

Table 1: The overall performance on the two datasets. All DP algorithms satisfy $(2, 10^{-9})$-DP. The underlined numbers indicate the best results over all methods; the bold numbers indicate the best results among the DP-based methods (See more results with larger $\varepsilon$ and $\delta$ in Appendix H).

Table 1 shows the performance on the two datasets. Pri-GPT is the GPT model trained on the private set $\mathcal{D}^{pri}$ without privacy protection, acting as an upper bound on the task performance. The high *P-n* scores on Pri-GPT show most generated n-grams may come from $\mathcal{D}^{pri}$, indicating a serious issue with privacy leaks in current text generation models. Pub-GPT uses the pre-trained GPT model to our setting without fine-tuning. Pub-GPT+$\tilde{\mathcal{D}}^{pub}$ is further fine-tuned on the pseudo data and outperforms Pub-GPT, indicating that the pseudo dataset helps in this task (Experiments in Appendix L also verify that.). The two baselines do not involve any information from the private data, so their *P-n* are the upper bound and their utilities are the worst (note that our goal is to do well on $\mathcal{D}^{pri}$ in inference).

DP learning algorithms are our main baselines. Pub-GPT+DP-SGD is initialized by Pub-GPT and further trained on $\mathcal{D}^{pri}$ with DP-SGD (Noisy-SGD) (Abadi et al., 2016; Kerrigan et al., 2020). Pub-GPT+DP-SGD+$\tilde{\mathcal{D}}^{pub}$ further fine-tunes the Pub-GPT+DP-SGD on $\tilde{\mathcal{D}}^{pub}$. The small gap between the two methods demonstrates DP-SGD cannot utilize $\tilde{\mathcal{D}}^{pub}$ well. Our method, SeqPATE, significantly outperforms the DP baselines (+39.3% and +28.0% in Bleu4) while ensuring the same strength of privacy protection in terms of *P-n* and $\varepsilon$. Note that, we conduct the experiments based on the sample level privacy of $\varepsilon = 2$ mentioned in Sec. 5.2, and the experimental results also reflect the protection on users' *secret phrase t* mentioned in Sec. 5.3. However, the strength of the protection is weaker than the sample-level protection. The model satisfies $(\varepsilon, \delta(\varepsilon, \tilde{s}_t))$-DP or $(s_t\varepsilon, \frac{e^{s_t\varepsilon}-1}{e^\varepsilon-1}\delta)$-DP on the secret phrase $t$ ($\varepsilon = 2$) as mentioned in Sec. 5.3. SeqPATE does better in empirical metrics on privacy compared to DP-SGD even if $\varepsilon = 2$ is the same theoretically; this provides empirical evidence of our discussion in Sec. 5.3 on providing stronger guarantees for phrases that appear more than once.

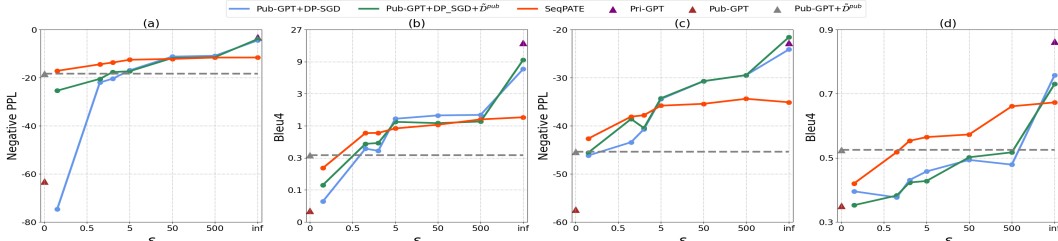

Figure 2: The private-utility tradeoff reflected in the Bleu-4 and negative PPL on a different $\varepsilon$.

The curves in Fig. 2 show the private-utility tradeoff of all DP algorithms in AirDialog (subfigure a & b) and Europarl_v6 (subfigure c & d) dataset [5]. DP with $\varepsilon \in [0.1, 5]$ is considered to provide a meaningful protection, and we observe that SeqPATE outperforms DP-SGD in this range. SeqPATE

---

[5]For non-DP methods, we consider $\varepsilon$ to be zero for baselines without using private data and $\varepsilon$ to be infinity for baselines without privacy protection.

does not work better than DP-SGD when $\varepsilon \geq 5$. The reason is DP-SGD approaches Pri-GPT as $\varepsilon$ approaches infinity (i.e. the noise approaches 0). However, SeqPATE with an infinite $\varepsilon$ is still weaker than Pri-GPT because each teacher in SeqPATE is trained on a small partition of the full data. The grey lines show the performance of Pub-GPT+$\tilde{\mathcal{D}}^{pri}$ and the area above the line indicates that the DP methods outperform Pub-GPT+$\tilde{\mathcal{D}}^{pri}$ (DP methods on the private data outperforms the non-DP methods on the non-private data). The area indicates a range where DP methods (with private data) are useful for this application. Our method always reaches that area when $\varepsilon \geq 1$ but DP-SGD hardly exceeds the grey line, the very trivial baseline, with $\varepsilon \leq 5$.

|  | AirDialog | | | Europarl_v6 | | |
|---|---|---|---|---|---|---|
|  | PPL | B-4 | P-4 | PPL | B-4 | P-4 |
| SeqPATE | 13.67 | 0.78 | 0.17 | 37.80 | 0.55 | 0.45 |
| −Merge P | 14.72 | 0.69 | 0.16 | 45.37 | 0.54 | 0.40 |
| −KL | 14.74 | 0.69 | 0.16 | 45.53 | 0.55 | 0.39 |
| −NLL | 13.60 | 0.74 | 0.18 | 38.09 | 0.54 | 0.45 |
| −Effi KD | 14.35 | 0.58 | 0.19 | 37.10 | 0.53 | 0.46 |
| −Gaussian | 13.69 | 0.60 | 0.16 | 38.24 | 0.50 | 0.43 |
| −All | 15.73 | 0.60 | 0.16 | 45.36 | 0.52 | 0.40 |

Table 2: Ablation studies.

|  | AirDialog | | | Europarl_v6 | | |
|---|---|---|---|---|---|---|
|  | PPL | B-4 | P-4 | PPL | B-4 | P-4 |
| top-$p$ | 13.67 | 0.78 | 0.17 | 37.80 | 0.55 | 0.45 |
| top-$k$=1 | 18.37 | 0.25 | 0.09 | 46.25 | 0.53 | 0.45 |
| top-$k$=10 | 14.57 | 0.79 | 0.21 | 45.90 | 0.55 | 0.40 |
| top-$k$=50 | 13.50 | 0.70 | 0.18 | 37.91 | 0.53 | 0.40 |
| top-$k$=100 | 13.96 | 0.76 | 0.19 | 38.48 | 0.57 | 0.50 |
| top-$k$=200 | 14.65 | 0.72 | 0.17 | 39.55 | 0.53 | 0.46 |

Table 3: Analyses about top-$k$ and top-$p$ strategies.

## 6.3 ABLATION STUDIES AND FURTHER ANALYSES

In the ablation studies (Table 2), SeqPATE mainly outperforms its variants shown in rows 2 to 7 verifying the effectiveness of our strategies, where $-X$ represents SeqPAET's variant without the strategy $X$. −Merge P is the variant aggregating the teachers by voting instead of merging the probabilities. Its poor performance shows that voting is not suitable for text generation with a large output space where the voting leads to low agreement rates. −KL means the student learns from the aggregation of teachers' top-1 predictions via NLL loss instead of KL loss. The sharp performance drop on −KL indicates the importance of KL loss. The performance of −NLL slightly decreases, indicating that NLL loss makes a little contribution to SeqPATE. The reason is that we have pre-trained on the student's training set via NLL before student training. The promotion caused by efficient knowledge distillation (Effi KD) on AirDialog is larger than that on Europarl_v6, which shows the "clever" student (e.g. models on AirDialog with low PPL and high Bleu) benefits more from this strategy since it can sharply save the privacy cost and spend the cost where the student strongly needs. −Gaussian means we use Laplace mechanism as mentioned in PATE instead of Gaussian mechanism, which shows that the Gaussian mechanism works better. −All indicates discarding all above strategies, which is similar to but not equivalent to the original PATE (the difference is that PATE needs to roll out all teachers (Sec. 4.1), which disables PATE from working on text generation). The poor performance of −All verifies that the original PATE is not adaptive to text generation.

Table 3 shows the performance of the top-$p$ and top-$k$ selection strategies (Sec. 4.3). As shown in row 1, our full model employs the top-$p$ selection (the threshold $\mu = 0.95$) surpasses most variants with manually chosen $k$ (rows 2 to 6). A selection with a too small $k$ ($k = 1$ & $k = 10$) implies discarding too much useful information from the supervision ($k = 1$ is different to $-$KL in Table 2, which uses Top-1 of teachers' results). A selection with oversize $k$ results in involving more useless noise: the candidates with very small probabilities should be rarely sampled during generation but random noise may increase their probabilities, so models may generate words that are misled by the noise. Supervisions on top-$k$ candidates with $k = 50$ and $k = 100$ yields relatively good performance.

Tables 4 & 5 in the Appendix shows more teachers lead to better results since the noise assigned to each teacher drops linearly as the teacher's numbers go up (as Eq. 2). We choose $\varepsilon = 2$ for all results in Table 2 to 5. P-n for all the results in a dataset to be roughly on the same level, except for some methods (e.g. row 2 in Table 3) that perform very badly in all metrics. It shows SeqPATE has the stable performance of protection if $\varepsilon$ is fixed (no matter which variant and strategy do we use).

## 7 RELATED WORK

Privacy protection is becoming crucial in deep learning models. Research in this area can be categorized into three directions (Liu et al., 2021). Encryption approaches are to add encryption on

training data (Brickell et al., 2007; Bost et al., 2015) or machine learning (ML) models (Aono et al., 2017). Aggregation approaches generally come along with distributed training where many parties join the training of an ML model with the purpose of maintaining the privacy of the dataset from each party (Shokri & Shmatikov, 2015; Konečný et al., 2016). Obfuscation approaches reduce the precision of the data or model by importing noise into the original input (Zhang et al., 2018), model parameters (Rubinstein et al., 2012), or model gradients (Abadi et al., 2016; McMahan et al., 2018).

As a sub-topic of the obfuscation approach, differential privacy (DP) (Dwork et al., 2006; 2014) formally defines and quantifies privacy. ML models with DP guarantee (Wang et al., 2015; Park et al., 2016; Ziller et al., 2021) prevents the existence of individual training examples from being detected (Carlini et al., 2019). Some researchers protect the privacy of empirical risk minimization (ERM) classifier (Chaudhuri et al., 2011) and SVM (Rubinstein et al., 2012) with DP. Following (Song et al., 2013), DP-SGD (Abadi et al., 2016) achieves DP on deep learning models by applying the noise to the clipped gradients. (Bu et al., 2020). Pichapati et al. (2019) adaptively clip the gradient on DP-SGD. PATE (Papernot et al., 2017) transfers the knowledge from teacher models trained on private data with injected noise to a student model. KNN-PATE (Zhu et al., 2020) refines PATE by accessing only the k-nearest neighbors from the private set instead of the whole private set. The above approaches are not customized for text generation models.

One direction of privacy protection in text generation is to protect author-level (user-level) information. The methods prevent attackers from inferring the author attributes (e.g. gender, age) (Li et al., 2018) and the relation between sensitive information and authors (Mireshghallah et al., 2021). Shokri et al. (2017) and Song & Shmatikov (2019) infer the membership (whether a given user's data is used to train the model) given a black-box model. Another direction is to prevent attackers from extracting sensitive information in the training set by analyzing the generated outputs (Nakamura et al., 2020; Kerrigan et al., 2020), which is an urgent need since Carlini et al. (2020) achieve the extraction. Our application focuses on this direction. In this direction, some researchers use regularization, including dropout (Srivastava et al., 2014) and quantization (Hubara et al., 2017), to restrict the model capacity to avoid models memorizing too much original text from the privacy training set (Carlini et al., 2019). Anonymization methods (Maeda et al., 2016; Suzuki et al., 2017) detects sensitive text spans in training sets and replace the spans with non-sensitive texts. The above methods are non-DP algorithms, which make sense intuitively but do not provide quantifiable and provable guarantees for privacy protection.

Some researchers apply DP to text generation. For user-level privacy, ER-AE (Bo et al., 2021) augments the semantic information in the generated text to hide authors' writing styles for authorship anonymization. McMahan et al. (2018) propose a recurrent language model with a DP guarantee against the identification of the users. Kerrigan et al. (2020) adapt DP-SGD to text generation, which prevents the (not the user-level) sensitive training data from being extracted from text generation models. Some of our concurrent works also follow this direction. Li et al. (2021) apply DP-SGD to large pre-trained language models and reduce the memory usage of those models. Yu et al. (2021) propose a framework to carry some advanced fine-tuning strategies on pre-trained language models with DP-SGD.

## 8 CONCLUSION

In this paper, we propose, SeqPATE, a framework to protect the privacy of the training data for text generation model with DP guarantees. SeqPATE achieves a good privacy-utility tradeoff by leveraging both private and public data. As an extension of PATE, SeqPATE can handle the sequential generation paradigm with large output space at each step and thus is adaptive for text generation models. We avoid rolling out the teachers by providing pseudo inputs for teacher inference and student training and aggregate teachers by merging probabilities instead of voting. We further reduce the output space by top-$k$ or top-$p$ selection and limit the privacy loss via an efficient knowledge distillation. SeqPATE improves the baselines a lot and makes the private data useful to text generation with a meaningful strength of privacy protection. In the future, we will apply SeqPATE to some real-world applications, e.g. dialog systems and writing assistants.

## 9 REPRODUCIBILITY STATEMENT

We've submitted our source code as the supplementary material. We introduce the two datasets we used in Appendix C, where we introduce the details about the dataset and also append the link or citation of the dataset. For all our experiments, we adopt *autodp* (Wang et al., 2019) — an open-source library that implements the analytical Gaussian mechanism for privacy accounting and calibration. The experimental setting and hyper-parameters can be found in Appendix D. Since we propose a new metric in our experiments, we introduce the new metric with detailed equations in Appendix E.

## 10 ETHICAL CONSIDERATIONS

This paper aims to protect privacy in training the text generation models, so our paper is designed for tackling some ethical issues about privacy concerns in some existing text generation models (e.g. GPT-2). In terms of motivation and the algorithm, our paper would not cause ethical issues.

However, we should also consider some special situations where someone intentionally applies our model to illegal applications. Someone may employ text generation models to create fake news or misinformation and protect himself or herself from being detected. For this purpose, they may use our model for their illegal application. In the future, we will add some constraints to our model so that our model cannot generate text for illegal applications (e.g. fake news).

In addition, we know large $\varepsilon$ (e.g. $\varepsilon = 500$) cannot provide a meaningful protection. We should carefully use our model and cannot assume the model is perfect no matter what parameters ($\varepsilon$ and $\delta$) do we use. One possible unethical application is to collect the data from users who believe our model can fully protect their privacy regardless of the strength of privacy protection (in terms of the value of $\varepsilon$). Hence, we kindly remind the researchers, who will use this model, to pay more attention to the strength of privacy protection. Further, we should prevent some researchers to collect the data from users who do not have a correct understanding of our algorithm, and the data collectors should make it clear that the potential risk in releasing data to our model.

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

## A  ALGORITHM FOR THE TRAINING OF SEQPATE

The pseudo codes of SeqPATE's training is shown in Algorithm 1.

---

**Algorithm 1** Training of SeqPATE

---

**Require:** $\mathcal{D}^{pri}, \mathcal{D}^{pub}$: datasets, $GPT$: a pre-trained GPT model.
1: $\{f_\phi^m\}_{m=1}^M$: $M$ teacher models, $f_\theta$: a student model, $f_\Theta$: a student model for self pre-training,
2: $\{\phi^m\}_{m=1}^M \leftarrow GPT, \Theta \leftarrow GPT$ # Initialize teachers and the student for self pre-training.
3: $GPT$ generates a pseudo dataset $\tilde{\mathcal{D}}^{pub}$ based on $\mathcal{D}^{pub}$.
4: $\{\mathcal{D}_m^{pri}\}_{m=1}^M \leftarrow \mathcal{D}^{pri}$ # Divide private dataset into $m$ subsets.
5: **for all** $m$ in $M$ **do**
6:   Train teacher $f_\phi^m$ on $\mathcal{D}_m^{pri}$
7: **end for**
8: Teachers $\{\phi^m\}_{m=1}^M$ conduct inference on $\tilde{\mathcal{D}}^{pub}$ to get $p_\phi^m(w_i \mid w_{1:i-1})$ required in Eq. 2 for all samples.
9:
10: Train $f_\Theta$ on $\tilde{\mathcal{D}}^{pub}$ # self pre-training for the student.
11: $\theta \leftarrow \Theta$ # Initialize the student model.
12:
13: **while** not converge **do**
14:   **for all** batch of samples $\{S\}^{\text{batchsize}}$ in $\tilde{\mathcal{D}}^{pub}$ **do**
15:     Student $f_\theta$ conducts feed-forward on $\{S\}^{\text{batchsize}}$.
16:     **for all** sample $S$ in the batch $\{S\}^{\text{batchsize}}$ **do**
17:       **for all** token $w_i$ in sample $S$ **do**
18:         $p_\phi(w_i \mid w_{1:i-1}) = \frac{1}{M} \sum_{m=1}^M p_\phi^m(w_i \mid w_{1:i-1})$ # Aggregate teachers' outputs
19:         Select only top-$k$ or top-$p$ predicted tokens as student's output.
20:         Obtain $\mathcal{L}_i^{\text{KL}}$ and $\mathcal{L}_i^{\text{NLL}}$ as Eq. 2. # Noise is added into $\mathcal{L}_i^{\text{KL}}$ to protect the privacy.
21:         Get $\mathcal{L}$ by combining $\mathcal{L}_i^{\text{KL}}$ and $\mathcal{L}_i^{\text{NLL}}$ # Efficient knowledge distillation.
22:       **end for**
23:     **end for**
24:     Update $\phi$ respect to $\mathcal{L}$.
25:   **end for**
26: **end while**

---

## B  DETAILED DEDUCTION OF THE SENSITIVITY IN SENTENCE LEVEL DP

We obtain Eq. 3 from Sec. 5.2.

$$\Delta_2^{(f)} = \|f(\mathcal{D}) - f(\mathcal{D}')\|_2 \le \|x_j - x_j'\|_2 = \left( \sum_{v=1}^{|\mathcal{V}|} (x_{jv} - x_{jv}')^2 \right)^{1/2} \tag{3}$$

We know $(x_{jv} - x_{jv}')^2$ is smaller than $|x_{jv} - x_{jv}'|$ since $|x_{jv} - x_{jv}'| \in (0,1)$ for each $v$. Hence, we have,

$$\left( \sum_{v=1}^{|\mathcal{V}|} (x_{jv} - x_{jv}')^2 \right)^{1/2} \le \left( \sum_{v=1}^{|\mathcal{V}|} |x_{jv} - x_{jv}'| \right)^{1/2} \le \left( \sum_{v=1}^{|\mathcal{V}|} |x_{jv} + x_{jv}'| \right)^{1/2}$$

We know $|a + b| = a + b$ when $a, b \in (0,1)$, so we have,

$$\left( \sum_{v=1}^{|\mathcal{V}|} |x_{jv} + x_{jv}'| \right)^{1/2} = \left( \sum_{v=1}^{|\mathcal{V}|} x_{jv} + \sum_{v=1}^{|\mathcal{V}|} x_{jv}' \right)^{1/2} = \left( 1 + 1 \right)^{1/2} \le \sqrt{2},$$

In summary, the upper bound of the sensitivity is,

$$\Delta_2^{(f)} = \|f(\mathcal{D}) - f(\mathcal{D}')\|_2 \le \|x_j - x_j'\|_2 = \sqrt{2},$$

## C  DETAILS ABOUT THE DATASETS

The AirDialog dataset (Wei et al., 2018) consists of 402,038 dialogues and each dialogue consists of more than two utterances. We treat each utterance as a sentence in our sentence completion task. We obtain the Europarl_v6 dataset from a machine translation benchmark [6] and we only use the monolingual English dataset with 2,015,440 raw sentences.

For the above datasets, we filter the short sentence with less than eight tokens. Then, the first four tokens act as the prefix, and the rest of the tokens acts as the output (ground-truth). We split each datasets into a private set $\mathcal{D}^{pri}$ and a public set $\mathcal{D}^{pub}$. For the AirDialog dataset, the private set contains 0.95M/5K/50K samples for training/validation/testing and the public set contains 40K/5K for training/validation. For the Europarl_v6 dataset, the private set contains 1.72M/10K/50K samples for training/validation/testing and the public set contains 40K/5K for training/validation. The vocabulary size for the two datasets is set to 50K and we replace the tokens out of the vocabulary with a special token.

## D  DETAILS ABOUT THE EXPERIMENTAL SETTING

All the comparing algorithms use the same base model, the GPT-small, which has 12 stacked layers as mentioned in the original paper (Radford et al., 2018). The pre-trained GPT model comes from the official website [7]. All the methods are initialized by the pre-trained GPT model. We used the Adam optimization scheme (Kingma & Ba, 2015) and we adjust the initial learning rate for all the comparing methods with a range of $0.01 \sim 0.00001$, which means we conducted a systematic search of the hyper-parameter space among all methods. We set the batch size as 32 for all the comparing baselines. We truncate the sentences with the maximal sentence length of 40. In the top-$p$ strategy, the threshold $\mu$ is 0.95. The factor $\lambda$ balancing the KL loss and NLL loss mentioned in Eq. 2 is 20.

## E  DETAILS ABOUT THE EVALUATION METRICS

To measure the quality of generation, PPL is a widely used metrics in text generation, especially language modeling, which measures the perplexity of generating the next token. B-n (Bleu-n) (Papineni et al., 2002) measures the degree of the n-gram matching between the generated results and the ground-truth text.

To measure the performance of privacy protection (we can also treat it as the privacy leak of the models), we aim to give an intuitive understanding of the protection on text generation scenario (even if the factor $\varepsilon$ in DP indicates the strength of privacy protection according to the DP theorem). Hence, we propose a new metric, P-N, indicating how many generated n-grams can be found in the private set $\mathcal{D}^{pri}$. We define $N_{\mathcal{Y}}$ as the set of unique n-grams in a generated sentence $\mathcal{Y}$ and define $N_{\mathcal{D}^{pri}}$ as the set of unique n-grams in the private data $\mathcal{D}^{pri}$. As shown in Eq. 4, P-N is the ratio of the n-grams that occurs in the two sets over n-grams occurs only in $N_{\mathcal{Y}}$ weighted by *idf*, where the *idf* is the inverse frequency of this n-gram indicating the sensitivity of this n-gram (Rare n-grams are likely to carry sensitive information, such as SSN number and address). P-N ranges from 0 to 1 and lower P-N indicates the more strong privacy protection (less privacy leak from the models).

For the choice of value N, we note that P-N with a very small value of N can hardly reflect the privacy loss since the uni-grams and bi-grams are very easy to be generated even if the model did not meet it from the private dataset $\mathcal{D}^{pri}$. For example, the N=1 does not make sense because it only reflects the vocabulary overlapping between two datasets. Thus, we choose 3 and 4 as the value of N.

---

[6]The description is in www.statmt.org/europarl; the data comes from statmt.org/wmt11/training-monolingual.tgz

[7]github.com/openai/gpt-2

|  | AirDialog | | | | | | |
|---|---|---|---|---|---|---|---|
|  | PPL | Blue1 | Blue2 | Blue3 | Blue4 | Pri3 | Pri4 |
| #teacher=1 | 19.66 | 8.52 | 2.48 | 0.84 | 0.30 | 0.10 | 0.04 |
| #teacher=10 | 18.29 | 8.27 | 2.38 | 0.81 | 0.29 | 0.14 | 0.06 |
| #teacher=200 | 16.86 | 9.01 | 2.70 | 0.96 | 0.33 | 0.16 | 0.07 |
| #teacher=2000 | 13.67 | 11.56 | 4.35 | 1.82 | 0.78 | 0.32 | 0.17 |

Table 4: Analysis about SeqPATE's performance with different teacher number on AirDialog dataset.

|  | Europarl_v6 | | | | | | |
|---|---|---|---|---|---|---|---|
|  | PPL | Blue1 | Blue2 | Blue3 | Blue4 | Pri3 | Pri4 |
| #teacher=1 | 44.64 | 12.33 | 3.64 | 1.11 | 0.40 | 0.56 | 0.32 |
| #teacher=10 | 44.22 | 12.67 | 3.73 | 1.16 | 0.43 | 0.58 | 0.33 |
| #teacher=200 | 43.55 | 13.10 | 4.06 | 1.29 | 0.49 | 0.60 | 0.36 |
| #teacher=2000 | 37.80 | 13.64 | 4.40 | 1.43 | 0.55 | 0.69 | 0.45 |

Table 5: Analysis about SeqPATE's performance with different teacher number on Europarl_v6 dataset.

$$N_{\mathcal{Y}} = \{\text{n-gram} \in \mathcal{Y}\}$$
$$N_{\mathcal{D}^{pri}} = \{\text{n-gram} \in \mathcal{D}^{pri}\}$$
$$\text{P-N} = \frac{\sum_{n \in N_{\mathcal{Y}} \cap N_{\mathcal{D}^{pri}}} idf_n}{\sum_{m \in N_{\mathcal{Y}}} idf_m} \tag{4}$$

## F  ANALYSES ABOUT THE NUMBER OF TEACHER MODELS

Table 4 and Table 5 show the analyses about SeqPATE's performance with different teacher number on our two datasets.

## G  THE COMPUTATIONAL COST OF SEQPATE

It seems that our model requires huge computational resources and a costly infrastructure to run. But in fact, our model is able to train and inference on a single GPU machine. In this section, we will introduce some simple strategies we used in our implementation and also introduce the total computational cost of our model.

**Memory usage and hard-disk space usage.**  Since our method uses a large number of teachers, the naive implementation of loading all teachers into memory for aggregation is impractical. However, note that our algorithm only needs to access each teacher's top-$k$ prediction. Therefore, we train teacher models sequentially. Once a teacher model is trained, we obtain its top-$k$ prediction ($k$=200 in our experiments) on the public training data and save the results (i.e. $k$ probabilities). Then, we discard the teacher model. Finally, SeqPATE only needs teacher supervision on a small number of samples. In our experiments, training on 1∼2k teacher labeled samples is sufficient. Overall, saving teachers' inference results uses 8∼16G. The memory usage is similar to that of a GPT2 model because we do not load all teacher models into memory and instead run inference sequentially and merge teachers' predictions offline.

**Training time.**  While we have a large number of teachers, each teacher is trained on only a small fraction of the entire dataset. Thus the time it takes to train all teachers is roughly equal to the time of training a single GPT2 model on the full dataset (of 1∼2M samples in our experiments).

In summary, with simple tricks, the teacher training and aggregation steps are not much more expensive than training a GPT2 model. Compared to standard NLG model training, our algorithm does not require special hardware or distributed learning.

## H    ADDITIONAL EXPERIMENTS ON LARGER $\varepsilon$ AND LARGER $\delta$

We adopted stronger privacy protection at $\varepsilon$=2 and $\delta$=1e-9 in Table 1, which results in a quite large utility gap between SeqPATE and the non-private training method (Pri-GPT) on the AirDialog dataset. Actually, our model performance can be increased a lot if we conduct weaker privacy protection (i.e. using a larger $\varepsilon$ and a larger $\delta$). In Table 6, we show the experimental results with $\varepsilon$=3 and $\delta$=1e-6 on AirDialog dataset.

| Methods | PPL | Blue1 | Blue2 | Blue3 | Blue4 |
|---|---|---|---|---|---|
| Pub-GPT+DP-SGD+$\tilde{\mathcal{D}}^{pub}$ ($\varepsilon$=2, $\delta$=1e-9) | 17.65 | 7.90 | 2.69 | 1.14 | 0.56 |
| Pub-GPT+DP-SGD+$\tilde{\mathcal{D}}^{pub}$ ($\varepsilon$=3, $\delta$=1e-6 ) | 13.87 | 10.24 | 4.95 | 2.83 | 1.74 |
| SeqPATE(Ours) ($\varepsilon$=2, $\delta$=1e-9) | 13.67 | 11.56 | 4.35 | 1.82 | 0.78 |
| SeqPATE(Ours) ($\varepsilon$=3, $\delta$=1e-6) | 9.94 | 15.55 | 7.82 | 4.43 | 2.54 |

Table 6: SeqPATE's performance with larger $\varepsilon$ and larger $\delta$ on AirDialog dataset.

## I    HANDLE LARGE $\varepsilon$: USING FEWER TEACHERS

As shown in Fig. 2, when we use large $\varepsilon$, our SeqPATE cannot obtain better performance than DP-SGD. As discussed in Sec. 6.2, the reason is that the amount of training samples for each teacher model is quite small in our model (especially, when the teacher number is large). We note that this can be improved by using fewer teachers. As shown in Table 7, we conduct experiments on AirDialog dataset with fewer (i.e. 100) teachers and larger $\varepsilon$ (i.e. 50 and 500), where SeqPATE outperforms DP-SGD. For DP-SGD, we choose Pub-GPT+ DP-SGD instead of Pub-GPT+DP-SGD+$\tilde{\mathcal{D}}^{pub}$ since Pub-GPT+ DP-SGD obtains higher performance when $\varepsilon$ is large.

| Methods | $\varepsilon$ | # teacher | PPL | Blue1 | Blue2 | Blue3 | Blue4 |
|---|---|---|---|---|---|---|---|
| Pub-GPT+DP-SGD | 50 | – | 11.21 | 9.68 | 4.45 | 2.43 | 1.43 |
| SeqPATE(Ours) | 50 | 100 | 10.03 | 10.50 | 4.66 | 2.54 | 1.47 |
| Pub-GPT+DP-SGD | 500 | – | 10.86 | 9.81 | 4.52 | 2.49 | 1.45 |
| SeqPATE(Ours) | 500 | 100 | 9.86 | 15.82 | 8.15 | 4.64 | 2.75 |

Table 7: To handle the large $\varepsilon$, using fewer teachers makes SeqPATE outperforms DP-SGD (on AirDialog dataset).

## J    EVALUATING PRIVACY LEAK IN ANOTHER METRIC (EXPOSURE)

We have evaluated the performance of privacy protection in P-3 and P-4. We also conduct experiments on another metric, *exposure*, to evaluate the privacy leak. The *exposure* is proposed by (Carlini et al., 2019). To calculate the score, we use three kinds of patterns "is <word> | are <word> | <digital>". The repetitions $r$ are 100 for all patterns, and the lengths of sampled words or digits are three. The candidate words are randomly sampled from the whole vocabulary and the digits are randomly sampled from $\{0, 1, ..., 9\}$. We calculate the average exposure score over the three kinds of patterns. We follow the implementation of the exposure score from [8].

| Methods | AirDialog | Europarl_v6 |
|---|---|---|
| Pri-GP | 3.13 | 2.83 |
| SeqPATE(Ours) | 0.91 | 1.55 |

Table 8: Results on *exposure* of PriGPT and SeqPATE on the two datasets.

Table 8 shows the experimental results, where the results are consistent with the results on P-3 and P-4 mentioned in Table 1. For example, in both metrics, the privacy leak in AirDialog is more serious than that in Europarl_v6 and the gap between Pri-GPT and SeqPATE on AirDialog is larger than that on Europarl_v6.

---

[8]github.com/tensorflow/privacy/tree/master/tensorflow_privacy/privacy

## K  Additional Experiments When the Dataset is Smaller

To verify our performance on smaller datasets, we produce a small dataset by randomly sampling 50k samples from the AirDialog dataset. The experimental results are shown in Table 9. In such a small dataset, our method still outperforms the DP-SGD with $\varepsilon = 2$ and $\delta = $ 1e-9.

| Methods | PPL | Blue1 | Blue2 | Blue3 | Blue4 |
|---|---|---|---|---|---|
| Pub-GPT+DP-SGD+$\tilde{\mathcal{D}}^{pub}$ | 25.93 | 7.64 | 2.26 | 0.87 | 0.23 |
| SeqPATE(Ours) | 16.68 | 9.48 | 2.96 | 1.08 | 0.39 |

Table 9: The performance on a small dataset (50k samples from AirDialog dataset). All the methods satisfy $(2, 10^9)$-DP.

.

## L  The Contribution of the Pseudo Public Dataset $\tilde{D}^{pub}$

We conducted experiments to verify the contribution of pseudo dataset $\tilde{D}^{pub}$ to our task on AirDialog dataset. The results in Table 10 shows that using $\tilde{D}^{pub}$ can prompt the performance a lot, where the promotion can be found in both SeqPATE and DP-SGD methods.

| Methods | Dataset | PPL | Blue1 | Blue2 | Blue3 | Blue4 |
|---|---|---|---|---|---|---|
| Pub-GPT+DP-SGD+$\tilde{\mathcal{D}}^{pub}$ | w/ $\tilde{\mathcal{D}}^{pub}$ | 17.65 | 7.90 | 2.69 | 1.14 | 0.56 |
| Pub-GPT+DP-SGD+$\tilde{\mathcal{D}}^{pub}$ | w/o $\tilde{\mathcal{D}}^{pub}$ | 18.47 | 7.17 | 1.95 | 0.66 | 0.24 |
| SeqPATE(Ours) | w/ $\tilde{\mathcal{D}}^{pub}$ | 13.67 | 11.56 | 4.35 | 1.82 | 0.78 |
| SeqPATE(Ours) | w/o $\tilde{\mathcal{D}}^{pub}$ | 18.09 | 7.96 | 2.26 | 0.78 | 0.28 |

Table 10: The comparison between using and not using the pseudo dataset, $\tilde{\mathcal{D}}^{pub}$.

## M  The Illustration of a Running Example.

Here, we will use an example to show our training processing. In this example, the prefix from the public dataset $\mathcal{D}^{pub}$ is "I want to book". We feed the prefix to pre-trained GPT2 to generate a pseudo sentence "I want to book a flight from Tokyo to Hawaii.". The pseudo sentence will serve as an example in the pseudo public dataset $\tilde{\mathcal{D}}^{pub}$. We feed the pseudo sentence to the teacher models to conduct the teacher inference and also feed it to the student model to conduct the feed-forward of the student training. Teacher models output the probability distributions on all words (10 words, in total) of the sentence. Then, we aggregate all teachers' probability distributions and add the calibrated noise on the aggregated distributions. The student model also generates the corresponding probability distributions on those words and we conduct the efficient knowledge distillation and top-$k$ or top-$p$ selection over the student's probability distributions. For example, if the student model can do well on the words ("I", "want", "book", "flight", and "Tokyo"), the student will only query the teachers' output distributions on the rest of words ("to", "from", "to", and "Hawaii") through the KL loss mentioned in 2. Besides, the student is always supervised by the NLL loss on the whole pseudo sentence ("I want to book a flight from Tokyo to Hawaii."). Finally, the student model conducts back-propagation according to the above losses.

