# OpenReview forum: "SeqPATE: Differentially Private Text Generation via Knowledge Distillation"
_ICLR.cc/2022/Conference — ICLR 2022 Submitted_

### Official Review · Reviewer_pnSu · 2021-11-06

**Correctness:** 4
**Technical Novelty And Significance:** 3
**Empirical Novelty And Significance:** 3
**Recommendation:** 8
**Confidence:** 4

**Main Review:**

Strengths: This paper is well-written and technically sound. Protecting text data is important given there is a rich body of work on attacks for such datasets; however, the protection technique that can provide provable DP guarantees is somewhat understudied. This is a timely and interesting topic and has many applications, for example, the smart compose. The approach is clearly stated, and the empirical evaluation is sufficient. In terms of novelty, SeqPATE has several non-trivial changes compared to the classical PATE.

Weaknesses: I have a few minor comments.

1. When \eps is large, DPSGD outperforms SeqPATE. In practice, the choice of \eps depends on the data, so could you offer any instructions on choosing the algorithm when \eps is not very small (like between 5-10, which is also common in practice)?

2. I'm confused about how you define P-n and why it can be considered as a privacy metric?

3. In Section 4.3, the sentence `` Eq. 1 shows the top-k selection at i-th step ...'' is confusing. Eq. 1 only shows the renormalization of the probabilities.


**Summary Of The Paper:**

This paper proposes SeqPATE that adapts PATE to text generation while satisfying differential privacy. To overcome the challenge of obtaining sequence-level supervision for text generation, it first generates pseudo inputs by completing the prefix using GPT-2 (a pre-trained language model) and thus reduces the problem to the next word prediction. Then it applies the PATE framework to privately train a student model with several algorithmic innovations, including (1) aggregating teachers' models by averaging their output distributions; (2) reducing output space by top-k/top-p selection; (3) efficient knowledge distillation by only querying the teachers' models when the student model performs poorly. (4) using both public labels and teachers' aggregation to supervise the learning. The paper gives privacy analysis and also shows it can offer privacy guarantees on user-level and secret n-grams. The paper also provides extensive experimental results and shows that it outperforms other DP learning algorithms when the \eps is small (<5). The experiments demonstrate the practicality of the algorithm.

**Summary Of The Review:**

Overall, I think it's a strong paper. The topic is timely and interesting. The writing is good, and the approach is clearly presented. The proposed approach contains several non-trivial changes compared to PATE. The paper provides extensive empirical evaluation, which demonstrates the practicality of SeqPATE.

---

> ### Author Response · Authors · 2021-11-20
> **Responses to reviewer4 (pnSu)**
>
> **Q1**: When $\varepsilon$ is large, DPSGD outperforms SeqPATE. Could you offer any instructions on choosing the algorithm when $\varepsilon$ is not very small (like 5-10)?
>
> We agree that the choice of privacy parameters is a policy decision users of our algorithm should provide.
>
> We have added experiments in weaker privacy regimes (larger $\varepsilon$ and larger $\delta$). **Our improvements over baselines are also significant** compared to $\varepsilon$ 0.1~5, if we use **fewer teacher** models. The table in Q4 in “common responses to all reviewers” shows the results with a larger $\varepsilon$. The first table in Q1 in “common responses to all reviewers” shows the results with a larger $\delta$. We note, however, that smaller $\varepsilon$ / $\delta$ is usually preferred for stronger and more theoretically meaningful privacy guarantees (e.g.  **$\varepsilon$ >5 says that an individual could be identified with confidence more than 99.33%** [Triastcyn et al., 2020]).
>
> As for choosing the algorithms, DPSGD-based algorithms are more suitable for larger $\varepsilon$ and PATE-based algorithms can do better with smaller $\varepsilon$. Please refer to the Q4 in “common responses to all reviewers” for further analysis and comparison between small and large $\varepsilon$.
>
> **Q2**: How to define P-N and why?
>
> **P-N is the fraction of unique n-grams in model generations that can be found in the private text**, which aims to reflect how much information in the **private text is leaked through the model generation outputs**. We also **evaluate with a new metric** ("exposure" [Carlini et al., 2019]) to measure the privacy leak. Please refer to the Q3 in “common responses for all reviewers” for the detailed explanation and the new experimental results.
>
> Q3: Typo in Section 4.3: `` Eq. 1 shows the top-k selection at i-th step ...''
> Sorry, this is a typo. I fixed it in the current version.
>
> [Triastcyn et al., 2020] Bayesian Differential Privacy for Machine Learning.

---

### Official Review · Reviewer_CVpe · 2021-11-08

**Correctness:** 3
**Technical Novelty And Significance:** 3
**Empirical Novelty And Significance:** 3
**Recommendation:** 6
**Confidence:** 3

**Main Review:**

## Strengths:
1. Training a DP language model is a less explored problem, and the authors provide a technically sound solution along with a well-setup empirical evaluation.
2. The paper is overall well written and the method is easy to follow.
3. The authors propose a new evaluation metric P-N to indicate how many generated n-grams can be found in the private set, which is an interesting and useful indicator in addition to PPL on the private set.
4. The authors also provide an interesting discussion on user-level DP and n-grams-level DP for language models.

## Weaknesses:
1. My main concern of this paper is whether SeqPATE can achieve satisfactory utility preservation given small epsilon (say epsilon=2). From Table 1, the B-4 score of SeqPATE is different from that of the upper bound by 2 orders of magnitude, which is a bit concerning. Moreover, the PPL gap between the upper bound and SeqPATE (13.67 - 3.88) is much larger than the gap between the lower bound and SeqPATE (19.39 - 13.67), which suggests that SeqPATE may barely learn useful information on the private data domain.
2. The advantage of SeqPATE vanished given a large epsilon compared with DP-SGD, which makes the algorithm less scalable. Although the authors explain it is due to the partitioning of the training data, could this issue be resolved if reducing the number of teachers?
3. The authors discuss user-level DP and n-grams-level DP. It is better to provide some quantitative evaluation on SeqPATE on these settings.
4. Some experimental setups are a bit unclear. it is better to include more experimental details in the main paper, e.g., how the hyper-parameters are chosen (say #/ teachers) and the computational overhead of SeqPATE and DP-SGD.

## Additional questions:
1. I see the paper is adopting a strict DP standard (delta=1e-9). Could raising the delta to 1e-6 or 1e-5 can help improve the utility of SeqPATE?
2. Why Pub-GPT + \tilde{D}^{pub} can significantly improve (the largest PPL improvement) the PPL of Pub-GPT, considering both methods use the public records only. This result is a bit weird and counterintuitive.

I am willing to raise my scores if the problems above can be well addressed.


**Summary Of The Paper:**

This paper proposes to protect the privacy of text generation models, which may leak sensitive information of the training data. Specifically, the authors propose SeqPATE, which applies the PATE framework to large-scale language models such as GPT. By reducing the output space and aggregating the supervision from teacher models, SeqPATE can help reduce the privacy cost. Empirical evaluation on two private text corpus shows that SeqPATE achieves a good trade-off between utility and privacy.


**Summary Of The Review:**

Strengths:
1. Training a DP language model is a less explored problem, and the authors provide a technically sound solution along with a well-setup empirical evaluation.
2. The paper is overall well written and the method is easy to follow.
3. The authors propose a new evaluation metric P-N to indicate how many generated n-grams can be found in the private set, which is an interesting and useful indicator in addition to PPL on the private set.
4. The authors also provide an interesting discussion on user-level DP and n-grams-level DP for language models.

Weaknesses:
1. My main concern of this paper is whether SeqPATE can achieve satisfactory utility preservation given small epsilon (say epsilon=2).
2. The advantage of SeqPATE vanished given a large epsilon compared with DP-SGD, which makes the algorithm less scalable.
3. The authors discuss user-level DP and n-grams-level DP. It is better to provide some quantitative evaluation on SeqPATE on these settings.
4. Some experimental setups are a bit unclear. it is better to include more experimental details in the main paper, e.g., how the hyper-parameters are chosen (say #/ teachers) and the computational overhead of SeqPATE and DP-SGD.

---

> ### Author Response · Authors · 2021-11-20
> **Responses to reviewer3 (CVpe)**
>
> **Q1**: SeqPATE may barely learn useful information on the private data domain.
>
> In general, the performance seems weak since we set a strong privacy protection ($\varepsilon$ = 2, $\delta$ = 1e-9). Following your advice, we relax the strength of private protection (using large $\delta$) and get better performance. The **results are shown in the first table of Q1** in “common responses to all reviewers”.
>
> There’re some other reasons resulting in the unsatisfactory performance, including the **ability of vanilla DPSGD and large utility loss since the model is suffering from a serious private leak**. Please refer to the Q1.2~Q1.4 in “common responses to all reviewers”.
>
>
> **Q2**: The advantage of SeqPATE vanished given a large epsilon compared with DPSGD.
>
>
> $\varepsilon$ = 0.1~5 provides meaningful privacy guarantees, where SeqPATE does well (e.g.  $\varepsilon$ > 5 says that an individual could be identified with a confidence more than 99.33% [Triastcyn et al., 2020]). Our new experiments show **SeqPATE also outperforms DPSGD with large $\varepsilon$ if we use fewer teachers**. Please refer to Q4 in “common responses to all reviewers”.
>
>
> **Q3**: Provide some quantitative evaluation on SeqPATE on user-level DP and n-grams-level DP.
>
> Even if experiments in Sec.6 are designed for the sentence-level DP, **the experimental results mentioned in Sec.6 can also serve as the quantitative evaluation of protection on user level and the user’s n-grams**. As mentioned in Sec.5.3, our method provides protection to secret n-grams (words or phrases), such as SSN numbers, phone numbers, and addresses. The secret n-grams carry sensitive information about individual users. So, one secret n-gram $t$ is very likely to come from a few users, where the user number is often 1 to 3 (e.g. parents and child who share the secret information). The secret n-gram $t$ also occurs very rarely in the dataset. We denote the frequency of $t$ as $s_t$, and denote $\tilde{s}_t$ = min{$s_t$, # users who know n-gram $t$}.
>
> As mentioned in Sec 5.3, if the teachers’ training data are partitioned by users, the experimental results in Sec.6 match the scenario where the algorithm **satisfies $(\varepsilon, \delta(\tilde{s}_t , \varepsilon))$-DP** to protect the secret n-gram $t$. Otherwise, the results in Sec.6 indicate the setting where the algorithm **satisfies $(s_t * \varepsilon, \frac{(e^{s_t  \varepsilon}-1)}{(e^{s_t}-1)} \delta)$-DP** to protect the secret n-gram $t$. We clarified it in Sec.5.3 and Sec.6.1. As mentioned in Sec. 5.3, SeqPATE can protect the secret phrases in user-level with few additional privacy loss, but DP-SGD cannot have such the merit.
>
>
> **Q4**: It is better to include more experimental details in the main paper, e.g., say #/ teachers.
>
> We introduce the experimental setups in the Appendix (e.g. teacher number is 2000). We’ve moved some important experimental setups in the main paper.
>
>
> **Q5**: The computational overhead of SeqPATE and DPSGD.
>
> We apply some strategies to save the computational cost on SeqPATE (mentioned in Q2 in “common responses to all reviewers”). As for the computational overhead of the teacher number, our model ran well with 2k teachers. If the teacher number is improved to 100k, the hard-disk usage will be 400~800G, which is too large for a single server. So, we can treat the computational overhead of the teacher number as 100k.
>
> For the batch size, in our setting, the overhead on DPSGD is 128; and overhead on SeqPATE is 384. (We are using V100 with 32G GPU memory).
>
> There’s no limitation for other hyper-parameters.
>
>
> **Q6**: Raise the $\delta$ to 1e-6 or 1e-5 to improve the utility.
>
> Thanks! We’ve followed your advice and the results are improved as your expectation. The results are shown in the first table of Q1 in “common responses to all reviewers”.
>
>
> **Q7**: Why does Pub-GPT + $\tilde{D}^{pub}$ significantly improve (the largest PPL improvement) the PPL of Pub-GPT?
>
> Pub-GPT is totally task-independent. However, $\tilde{D}^{pub}$ is a dataset carring the information about the target task, since $\tilde{D}^{pub}$ is generated by GPT given the public dataset $D^{pub}$, which consists of sentence prefixes of samples in the target task. So, **Pub-GPT + $\tilde{D}^{pub}$ benefits from the task-related information in $\tilde{D}^{pub}$**.
>
> [Triastcyn et al., 2020] Bayesian Differential Privacy for Machine Learning.

---

> > ### Comment · Reviewer_CVpe · 2021-11-24
> > **Thanks for the response**
> >
> > Thank the authors for the response! I also read reviews from other reviewers as well as the corresponding responses. The answer clears most of my concerns. The new experimental results look convincing to me.
> >
> > However, I have an additional question: while this paper focuses mainly on the semi-supervised learning setting that assumes that there is a public dataset from the target domain, I am wondering if the paper can be adapted to the setting where only the pre-training data is available and no public dataset from the target domain is available. If so, what is the advantage of SeqPATE over DPSGD-based methods? I am asking because this setting is especially practical for language models, which have access to the public data for pre-training and the sensitive private data for domain-specific fine-tuning.

---

> > > ### Author Response · Authors · 2021-11-28
> > > **Responses to the further questions from reviewer 3 (CVpe)**
> > >
> > > Thank you so much for your positive comments.
> > >
> > > **Q1**: Can SeqPATE work without the public dataset from the target domain?
> > >
> > > PATE does require unlabeled public data to train the student model, but there are ways to get around the problem of lacking public training data. In the paper, we assume that the public in-domain data comes from consented users or is collected separately, which is a reasonable assumption in practice.
> > >
> > > If even such data is not available, we can 1) select samples similar to the target domain from the public pre-training dataset as our public set; or 2) synthesize target-domain samples via template-based approaches. Note that SeqPATE only requires a small public set (100~500 samples) so the **cost of obtaining it is small**.
> > >
> > > In addition, we would like to highlight that it’s also important to **protect the privacy of pre-training data** (although it’s often considered public). Carlini et al. [2019] show that attackers can extract private information such as phone numbers from public pre-training data as well. Accordingly, our results focus on training private language models on large data (as opposed to the “pre-train then fine-tune” setting).
> > >
> > > **Q2**: The advantage of SeqPATE over DP-SGD-based methods in the new settings.
> > >
> > > If the public set is totally independent from the target domain, SeqPATE hardly has the advantage over DP-SGD if there's a huge domain shift between pre-training data and the target domain.
> > >
> > > In general, the advantage of SeqPATE over DP-SGD are as follows,
> > >
> > > a) SeqPATE is a **model-agnostic** framework, thus can be easily adapted to non-gradient-based NLP methods (e.g. retrieval-based systems and in-context learning), where SGD is not applicable. Also, different from DP-SGD, **SeqPATE’s privacy loss will not increase as the model becomes large.**
> > >
> > > b) SeqPATE achieves **user-level privacy on secret phrases** without additional privacy cost by training each teacher on data from a single user, whereas DPSGD incurs a privacy cost that grows linearly with the number of occurrences of the secret phrase (as mentioned in Sec. 5.3).
> > >
> > > c) SeqPATE achieves higher utility with strong privacy protection ($\varepsilon \leq 2$).
> > >
> > >
> > > [Devlin et al., 2019] BERT: Pre-training of Deep Bidirectional Transformers for Language Understanding.
> > >
> > > [Carlini et al, 2019] The secret sharer: Evaluating and testing unintended memorization in neural networks.

---

> > > > ### Author Response · Authors · 2021-12-07
> > > > **Thank you, and any more concerns? (To reviewer3 (CVpe))**
> > > >
> > > > Thank you very much for your positive comments and suggestions!
> > > >
> > > > **As you mentioned, we've clarified most of your concerns, and the new experiments look convincing.** May I know if there is anything else we need to address and make the paper more satisfactory to you? I am willing to solve any concern or problem.

---

> > > ### Author Response · Authors · 2021-11-29
> > > **Additional Experimental Results about Protecting Users’ Secret Phrases**
> > >
> > > According to your kind suggestion, we conducted additional experiments to show SeqPATE's advantage in protecting users’ secret phrases. **The results show that SeqPATE can provide much stronger protection than DP-SGD (in terms of $\varepsilon$ for users’ secret phrases**, even if SeqPATE achieves a better utility. In the experiments, we treat the “full name of the user” as the secret phrase we want to protect. And the teachers’ training data are partitioned by users (each user’s samples occur in only one teacher model), which is the only difference between the SeqPATE mentioned in our paper. Thank you very much for your suggestion and positive comments.
> > >
> > > |                     | $\varepsilon$ at sample level | $\varepsilon$ for users' secret phrases | PPL   | BLEU-1 | BLEU-2 | BLEU-3 | BLEU-4 |
> > > | ------------------- | ------------------------------  |-----------------------------------------------  | ----- | ------ | ------ | ------ | ------ |
> > > | DPSGD | 3 | $\textbf{123}$ |13.87 | 10.24  | 4.95   | 2.83   | 1.74 |
> > > | SeqPATE(ours) | 3 | $\textbf{3}$ | 11.45 | 12.83 | 6.19   | 3.33   | 1.86   |

---

### Official Review · Reviewer_wFLx · 2021-11-09

**Correctness:** 4
**Technical Novelty And Significance:** 2
**Empirical Novelty And Significance:** 2
**Recommendation:** 3
**Confidence:** 4

**Main Review:**

This paper provides results in the very interesting direction of private language models. I believe that, given the recent progress in non-private language models, the time is ripe to further investigate this area privately, which has been sorely under-explored. As a result, I commend the authors for taking steps in this direction.

First, I will evaluate the algorithmic contributions. As I enumerated above, there are a few modifications to the basic PATE setting. Of these, I believe using the student model to reduce the support size is an interesting idea. Using a pre-trained GPT-2 model to extend the training data is also kind of nice, though it is not 100% clear why this is necessary in the experimental settings considered. In the example provided, I understand: "Cats sit" is too short a sentence to perform meaningful training from. However, the datasets used seem to have much larger utterances (indeed, it says in Appendix C that sentences less than 8 tokens long were filtered out). The other two modifications seem relatively straightforward: it is of course natural to consider using multiple prefixes of the same pseudo sentence, rather than performing predictions sequentially. And performing PATE on the output distribution rather than the output prediction is also natural (I would be surprised if it has not been done before). I believe it may also be inspired directly from knowledge distillation (which is uncited and uncredited). Overall, the algorithmic modifications are nice, but nothing offering significantly new insights into the nature of natural language generation.

Next, I will comment on the experimental results. The utility metrics provided in Table 1 seem to indicate that the proposed method performs somewhat better than the previous methods which are private with respect to the sensitive dataset. However, the utility is still very far from the non-private utility. Perhaps this is to be expected, given the general cost of guaranteeing DP, but other recent results seem to show how effective pre-trained models are in terms of maintaining utility even with DP -- see, for example, Tramer-Boneh (https://arxiv.org/abs/2011.11660), which demonstrates the efficacy of transfer learning to image classification (resulting in relatively comparably small drops to the accuracy versus no transfer learning), as well as two recent arXiv papers on the same topic (https://arxiv.org/abs/2110.06500 and https://arxiv.org/abs/2110.05679) which achieve relatively small loss in utility compared to the non-private pre-trained model. Of course the latter two works appeared only after the deadline and thus were impossible to compare with, so perhaps one should disregard them, but the story of both papers seem to be at odds, which is puzzling to me. Regardless, the [TB20] paper still indicates the significant power of transfer learning for private learning). Additionally, I noticed that experiments were only performed on two relatively large datasets. This begs the question of what happens when the datasets are relatively small. Lack of accuracy on smaller datasets may be why the authors did not evaluate on more common GEM benchmarks. Comparison with prior methods seem a little mixed. While SeqPATE indeed outperforms previous methods in various regimes, it is somewhat inconclusive at eps = 5, which is closer to the regime usually considered in DP ML papers. As a final note, my impression is that this is an incredibly costly infrastructure to run. I believe this uses k = 2000 teachers, each of which is GPT-2 Small, resulting in an overall model which is comparable in number of parameters to the full GPT-3. This sounds impractical to run in most settings, and this matter is not discussed at all in the paper. Overall, the experimental results are modest, but given this discussion, I am not sure if they mark a qualitatively impressive shift in our understanding of the problem (which, as I mentioned before, is seemingly quite under-explored).

My final major comment is on the writing. The quality of the language is rather poor, with issues in almost every sentence, to the effect that the final paper feels very rushed. I tried to be charitable to the authors and fill in the gaps intelligently, but there were certain sentences which were incomprehensible in their current state. Beyond this, other (comparatively minor) writing issues include insufficient discussion of related work, incorrect citations, and unusual choice of content in the body. I list some of these issues below. Overall, presentation issues in the submitted version are to the level where I would not recommend anyone to read this until it is better polished.

Further points and comments:
- The citation for differential privacy is Dwork, McSherry, Nissim, and Smith, TCC 2006.
- Bassily, Smith, Thakurta, FOCS 2014 should be cited as another reference for DPSGD.
- As mentioned above, what happens if one just uses D^pub instead of ~D^pub? Why is this insufficient for the datasets used in the experiments? This question is true for many of the approaches studied, not just SeqPATE.
- The term knowledge distillation (KD) is used extensively in the paper. As I am sure the authors are aware, KD refers to a specific technique for training from a teacher. Is the KD used in the paper the same as KD as used in the literature? If so, then it should be cited and discussed (or otherwise, contrasted with). If it is different, then the term KD should be changed.
- It may be helpful for the presentation to keep going with the running example demonstrated in Figure 1, and how the procedure would work with this concrete instance.
- Although I am not too concerned with what appears in the body of the paper versus the appendix, some choices that I found unusual include: basics of DP and Equation (1), which may be immediate or already known to most readers.
- It is not clear what bold and underlined numbers in Table 1 are meant to represent.
- I am curious what happens for the non SeqPATE methods if one uses D^pub instead of ~D^pub. It feels like you're cheating if you're just using the model to generate more training data, and it may or may not have an effect.
- The choice of P-3 or P-4 is not clear as a privacy metric. It seems to say something more about the nature of the dataset. See in particular how these numbers differ significantly between AirDialog and Europarl_v6. The canary approach used in the Carlini et al Secret Sharer paper seems more meaningful to me.
- Broken reference in Appendix D
- It appears that footnote 7 was never filled out appropriately
- It seems like more teachers result in monotonically better utility. What is the limit of this phenomenon?


**Summary Of The Paper:**

This paper proposes a variant of PATE (from ICLR 2018), a method intended for standard supervised learning tasks. The present work instead focuses on natural language generation, an inherently sequential task, which introduces new challenges. Relative to PATE (and assuming knowledge of it), some differences in the method involve: extending the public data sentences into "pseudo sentences" using GPT-2, simultaneously performing training on all prefixes of the pseudo sentences, averaging the teacher model output distributions rather than aggregating their max score (i.e., their prediction), and using the student model to reduce the support size for said output distributions. Experimental evaluation is provided for two datasets, AirDialog and Europarl_v6. Perplexity and BLEU scores improve upon those which do not use the private data, and those which use previous methods with the private data.


**Summary Of The Review:**

A nice direction for investigation. But the approach and findings don't appear significant enough to warrant acceptance in ICLR right now. The paper is further hampered by the very poor presentation.

---

> ### Author Response · Authors · 2021-11-20
> **Responses to reviewer2 (wFLx) [Part1]**
>
> **Q1**: Example in Fig1 ("Cats sit"..) is too short.
>
> This is mainly for illustration. In our experiments, the prefix length is 4, and the length of the whole sentence is at least 8.
>
>
> **Q2**: Using output distribution rather than output prediction is natural.
>
> For the classification tasks (the candidate size is not large), there's a little difference between using output distribution and output prediction since voting on the prediction can estimate the merged distribution. So, most PATE-based models for classification use the output prediction. In our work, we adapt PATE to natural language generation (NLG), where merging teachers with output distribution is crucial for the large search space in NLG, even if merging teachers with output distribution is not so hard in PATE.
>
>
> **Q3**: Missing citation on knowledge distillation (KD) papers when mentioning merging output distribution.
>
> Thanks for pointing this out. Merging output distribution is indeed also used in KD in NLG and we have cited KD papers in the new version.
>
>
> **Q4**: Our approach is straightforward and doesn’t add new insights into NLG.
>
> 1. Our paper focuses more on practical applications of (existing) DP learning algorithms to NLG rather than novel methods. Thus the main contribution is to **adapt PATE to NLG problems with simple and effective techniques** (similar to the other two submissions that make DPSGD practical on NLP problems).
>
> 2. In addition, we believe exploring PATE adds new knowledge to this (underexplored) field. Compared to DPSGD, PATE is **model-agnostic**, thus can be applied to methods that **do not use gradient-based optimization**, such as retrieval-based methods and in-context learning; both are increasingly popular in NLP. Further, our results show that SeqPATE provides better performance **under strong privacy protection (i.e. small $\varepsilon$)**. We also refer the reviewer to Q1.4 in the common response.
>
>
> **Q5**: Considering two recent papers (ICLR submissions) about DPSGD for NLG, the utility seems far from the non-private utility.
>
> We’d also like to point out that under similar settings (i.e. same privacy levels and vanilla implementation of DPSGD) the results in our paper are consistent with observations in the two submissions. Please refer to the Q1 in “common responses for all reviewers”
>
>
> **Q6**: [TB20] (Tramer et al., 2020) indicates the power of transfer learning for private learning.
>
> 1. The three recent papers [Yu et al., 2021a; Li et al., 2021; Tramer et al., 2020] mainly focus on transfer learning (i.e. fine-tuning a pre-trained model). **Our setting is not particularly focused on transfer learning**. We are interested in training an NLG model from larger corpora (possibly from scratch). Our task is more similar to machine translation or conversation systems, which are typically trained on large datasets and can work well without pre-training. We conducted the experiments with pre-trained models to match state-of-the-art results.
>
> 2. As for the **specific paper**, [Tramer et al., 2020] focuses on image classification tasks, which is a more mature task for DP algorithms since the original DPSGD [Abad et al., 2016]. There’re many recent techniques designed for applying DP to this task such as Scattering Network [Tramer et al., 2020]. And the number of parameters of GPT2-based (117M $\sim$ 1542M) NLG models is much larger than that of image classification models (26K $\sim$ 168K). Large parameters increase the difficulty of applying DPSGD [Yu et al., 2021b].
>
>
> **Q7**: Why not use small datasets (e.g. datasets in GEM)?
>
> 1. We produce a small dataset by randomly sampling 50k samples from the AirDialog dataset. The experimental results are as follows and attached to our Appendix. Our method still outperforms the DPSGD with $\varepsilon$=2 and $\delta$=1e-9.
>
>     |                   | PPL   | BLEU-1 | BLEU-2 | BLEU-3 | BLEU-4 |
>     | ----------------- | ----- | ------ | ------ | ------ | ------ |
>     | DPSGD             | 25.93 | 7.64   | 2.26   | 0.87   | 0.23  |
>     | SeqPATE(ours) | 16.68 | 9.48   | 2.96   | 1.08   | 0.39   |
>
> 2. As mentioned in the answer to Q6, unlike the setting in [Yu et al., 2021a; Li et al., 2021], which benefits from the pre-trained model, our setting (sentence completion) is more similar to machine translation (MT) and conversation systems, which are **typically trained on large datasets** (MT: 0.5M $\sim$ 50M [WMT], conversation: 1 $\sim$10M [Lowe et al., 2016]) and work well even without pre-training.
>
> 3. **Some tasks (e.g. table/concept/triple-to-text generation) does not require large datasets**. We say they are easy because in their tasks the major output words come from the input and the structure of output texts can follow some prototypes or templates. In GEM, many small dataset is on table/concept/triple-to-text task (e.g. CommonGEN, DART, E2E, ToTTo, and WebNLG). Compared to those tasks, our task, MT, summarization tasks are more challenging thus need more samples.

---

> > ### Author Response · Authors · 2021-11-20
> > **Responses to reviewer2 (wFLx) [Part2]**
> >
> > **Q8**: The advantage of SeqPATE vanishes at $\varepsilon$ = 5.
> >
> > $\varepsilon$ = 0.1~5 provides meaningful privacy guarantees, where SeqPATE does well (e.g.  $\varepsilon$ > 5 says that an individual could be identified with confidence more than 99.33% [Triastcyn et al., 2020]). Our new experiments show SeqPATE also outperforms DPSGD with **large $\varepsilon$ if we use fewer teachers**. Please refer to Q4 in “common responses to all reviewers”.
> >
> >
> > **Q9**: The number of the model parameters sounds too many to run practically.
> >
> > With simple tricks, our model training is **not much more expensive than training a GPT2 model** in the memory/hard-disk usage and the training time. Please refer to Q2 in “common responses to all reviewers”.
> >
> > **Q10**: The weakness of our paper writing.
> >
> > We start to refine the paper writing in the rebuttal period and have updated a new version on Nov. 22. After the rebuttal period, we will also continue to polish the paper.
> >
> >
> > **Q11**: The wrong citation of DP on [Dwork et al., 2006] and [Bassily et al., 2014].
> >
> > Thanks. We cited the Dwork and Roth textbook for the ($\varepsilon$,$\delta$)-DP definition. It is a good point you raised that we should’ve cited the original DP paper “Calibrating noise to ...” too. It’s fixed in the paper now.  We also added [Song et al.,  Bassily et al; and Abadi et al.] when we first introduce the NoisySGD mechanism.
> >
> >
> > **Q12**: Only uses $D^{pub}$ instead of $\tilde{D}^{pub}$ (for both SeqPATE and non-SeqPATE model)? Why $D^{pub}$ is insufficient for the datasets?
> >
> > In many real scenarios, most users hesitate to contribute their data to such a public dataset since algorithms would make full use of the public dataset without catering to its privacy. The public data may come from volunteers or paid annotators. So, the public data is usually insufficient and expensive.
> >
> > To solve that issue, there are two possible ways: one is to train on the insufficient original data $D^{pub}$, another is to extend the original dataset and train on the extended dataset $\tilde{D}^{pub}$. According to your kind suggestions, we conducted experiments to compare the two strategies and the result shows the **latter one (using $\tilde{D}^{pub}$) is better**. The following table shows the results, where we verify it on **both SeqPATE (ours) and non-SeqPATE (DPSGD) methods**. We've added the experiments to the Appendix of our paper.
> >
> > |                     |                   | PPL   | BLEU-1 | BLEU-2 | BLEU-3 | BLEU-4 |
> > | ------------------- | ----------------- | ----- | ------ | ------ | ------ | ------ |
> > | DPSGD | w/ $\tilde{D}^{pub}$ | 17.65 | 7.90 | 2.69 | 1.14 | 0.56 |
> > | DPSGD | w/o $\tilde{D}^{pub}$ | 18.47 | 7.17   | 1.95   | 0.66   | 0.24   |
> > | SeqPATE(ours) | w/ $\tilde{D}^{pub}$ | 13.67 | 11.56  | 4.35   | 1.82   | 0.78   |
> > | SeqPATE(ours) | w/o $\tilde{D}^{pub}$ | 18.09 | 7.96   | 2.26   | 0.78   | 0.28   |
> >
> >
> > **Q13**: The meaning of knowledge distillation (KD).
> >
> > Yes. As your understanding, knowledge distillation (KD) is equal to the teacher-student framework where the student is supervised by the teacher, which is also mentioned in [Chen et al., 2020]. We have explained it and added citations in the current version.
> >
> >
> > **Q14**: Present the running example demonstrated in Figure 1.
> >
> > We will explain the procedure with the example in Figure 1. We have attached it to our new version.
> >
> >
> > **Q15**: Basics of DP and Equation (1) should be in the appendix.
> >
> > We tried to provide rigorous definitions needed for stating our results, as well as narratives to ensure the paper is a good read for the ICLR readers. DP is still a foreign concept to most NLP researchers.
> >
> > Moreover, we would like to emphasize that Lemma 4 and 5 represent the state-of-the-art in privacy accounting as they give tighter privacy loss computation than moments accountant (or RDP).  The reason why we can leverage Lemma 4 (Analytical Gaussian Mechanism) and Lemma 5 (GDP composition) is that our methods use only Gaussian mechanisms, while DPSGD critically relies on “amplification by sampling” for which the best available tool is RDP to our knowledge.
> >
> >
> > **Q16**: The choice of P-3 or P-4 is not clear as a privacy metric.
> >
> > P-N is the fraction of unique n-grams in model generations that can be found in the private text, which aims to reflect how much information in the private text is leaked through the model generation outputs. We also **evaluate with a new metric ("exposure" [Carlini et al., 2019])** to measure the privacy leak. Please refer to the Q3 in “common responses for all reviewers” for the detailed explanation and the new experimental results.
> >
> >
> > **Q17**: What’s the meaning of bold and underlined numbers in Table 1.
> >
> > The underlined numbers indicate the best results over all methods; the bold numbers indicate the best results among DP-based methods. We’ve added the explanation to the table caption.

---

> > > ### Author Response · Authors · 2021-11-20
> > > **Responses to reviewer2 (wFLx) [Part3]**
> > >
> > > **Q18**: More teachers result in monotonically better utility. What is the limit of this phenomenon?
> > >
> > > As the computational overhead mentioned in Q2 of “common responses to all reviewers”, if the teacher number is improved to 100k, the hard-disk usage will be 400 $\sim$ 800G, which is too large for a single server. So, we can treat the computational overhead of the teacher number as 100k.
> > >
> > > In our setting, more teachers lead to better performance, when the teacher number is no more than 2000. But, **using more teachers also leads to fewer training samples for each teacher**, which also reduces the performance by weakening the teacher models. So, more teachers cannot always increase the performance and the optimal number of teachers depends on the particular settings. This theoretical analysis of PATE [Bassily et al, 2018; Liu et al, 2021] made the dependence on the number of teachers explicit (TL;DR: it needs to be chosen carefully, **cannot be too large or too small**).
> > >
> > >
> > > **Q19**: The typos.
> > >
> > > Thanks for pointing out the typos and we’ve fixed them in the updated paper.
> > >
> > > [Li et al., 2021] Large Language Models Can Be Strong Differentially Private Learners.
> > >
> > > [Yu et al., 2021a] Differentially Private Fine-tuning of Language Models.
> > >
> > > [Tramèr et al., 2021] Differentially Private Learning Needs Better Features (or Much More Data).
> > >
> > > [Yu et al., 2021b] Do Not Let Privacy Overbill Utility: Gradient Embedding Perturbation for Private Learning.
> > >
> > > [Abadi et al., 2016] Deep Learning with Differential Privacy.
> > >
> > > [WMT] https://www.statmt.org/
> > >
> > > [Lowe et al., 2016] The Ubuntu Dialogue Corpus: A Large Dataset for Research in Unstructured Multi-Turn Dialogue Systems.
> > >
> > > [Bassily et al, 2018]  Model-agnostic private learning via stability.
> > >
> > > [Liu et al, 2021] Revisiting Model-Agnostic Private Learning: Faster Rates and Active Learning.
> > >
> > > [Chen et al., 2020] ​​Distilling Knowledge Learned in BERT for Text Generation.

---

> ### Author Response · Authors · 2021-11-30
> **Thanks for your comments!**
>
> Thank you very much for your comments and suggestions!
>
> Did our explanation and new experiments resolve your concerns? We will keep the discussion forum open throughout the meta-review period in order to accommodate any last-minute discussions for such cases. If the discussion forum is not available for the authors, we will respond on our anonymous Github (github.com/anonymityForPaperSubmit123/ICLR22SeqPATE).

---

### Official Review · Reviewer_386M · 2021-11-09

**Correctness:** 3
**Technical Novelty And Significance:** 2
**Empirical Novelty And Significance:** 3
**Recommendation:** 6
**Confidence:** 4

**Main Review:**

Strengths: I think the paper does a good job of adapting PATE to the text generation framework and making it work reasonably well. Given that PATE is a very general framework that works for any learning algorithm and not just deep learning, it is useful to have adaptations of it to sequence/text generation settings.

Weaknesses:

I am not fully convinced by the authors claim that their adaptation of PATE beats DPSGD.

(1) The authors cite Kerrigan et al (2020) to claim that DPSGD doesn't work for text-generation using large models like GPT2 due to high dimensionality of the parameter space. This is debunked conclusively in the two recent papers by Li et al 2021 (https://arxiv.org/abs/2110.05679) and Yu et al 2021 (https://arxiv.org/abs/2110.06500). Li et al (2021) show that by careful choice of hyperparameters, training pretrained GPT2 on private datasets using DPSGD achieves performance comparable to non-private training. We also note that these are very recent and possibly concurrent with this work. But nevertheless, the authors should not be coming to an opposite conclusion as these papers.

(2) I am also not convinced by the experiments in Table 1. The authors compare their approach to the baseline DPSGD+$\tilde{D}^{pub}$. This means that, from what I understand, a model is trained from scratch on private data $D^{priv}$ using DPSGD and then fine-tuned on $\tilde{D}^{pub}$. I think that the correct baseline to compete against is Pub-GPT + $\tilde{D}^{pub}$ + DPSGD on $D^{priv}$ in that order. That is we are fine-tuning the pretrained GPT2 on $\tilde{D}^{pub}$ and then fine-tuning on $D^{priv}$ using DPSGD. I will be more convinced that their framework works if they compare against this benchmark.

(3) Also to claim that, their method is beating DPSGD, they have to demonstrate they did a systematic search of the hyperparameter space for DPSGD. But I didn't find any evidence of this in the paper. The importance of choosing right hyperparameters for DPSGD is demonstrated in the work of Li et al (2021).

Other remarks: Is the grey line in Figure 2, Pub-GPT + $\tilde{D}^{pub}$? Is it a typo?


**Summary Of The Paper:**

This paper tries to adapt the PATE (Private Aggregation of Teacher Ensembles) framework for DP training of machine learning models to the setting of text generation. There are some significant challenges in adapting PATE to this framework.  PATE as originally proposed is designed to work for classification tasks with a small number of labels. Text generation or next word prediction has too many labels and a direct application of PATE doesn't work. This paper solves this issue by aggregating only top-k predictions of each teacher where the top-k is computed using the prediction of the student on public data and so doesn't leak extra privacy. The paper also crucially uses public data and large pretrained models like GPT2 to further improve the performance of DP learning using PATE on private data.

They claim that their approach improves upon DPSGD/Noisy SGD (which is the standard DP algorithm for deep learning) significantly on some standard benchmarks.

**Summary Of The Review:**

Because I am not fully convinced the by the claims of the authors, I rate it marginally below acceptance threshold. I am willing to update my score based on further evidence by the authors in support of their claims.

---

> ### Author Response · Authors · 2021-11-20
> **Responses to reviewer1 (386M)**
>
> Q1: GPT2 with DPSGD on text generation cannot work so well due to high dimensionality, which seems to be opposed to the other two ICLR22 submissions.
>
> Thanks for pointing out the two concurrent papers. We have corrected the statement in the revision given the new evidence.
>
>   We’d also like to point out that under similar settings (i.e. same privacy levels and vanilla implementation of DPSGD) the results in our paper are consistent with observations in the two submissions. Please refer to Q1 of “common responses to all reviewers”.
>
>
> Q2: Missing baseline: compare against Pub-GPT + DPSGD + $\tilde{D}^{pub}$ on $D^{priv}$.
>
>   The proposed setting is indeed the setting used in our experiments - we have made this more clear in the current version. Specifically, DPSGD + $\tilde{D}^{pub}$ (row 5 in table 1) refers to the setting of “Pub-GPT + DPSGD + $\tilde{D}^{pub}$ on $D^{priv}$.”, where we fine-tune the pre-trained GPT2 model on $\tilde{D}^{pub}$ without noise, and then train it on $D^{priv}$ with DPSGD. We have changed the name of this experiment to “Pub-GPT + DPSGD + $\tilde{D}^{pub}$” as suggested.
>
>
> Q3  Did you conduct a systematic search of the hyperparameter space for DPSGD?
>
>   Yes, we described hyperparameters search for DPSGD in Appendix D. For the choice of the experimental settings, we did not tune batch size, vocabulary size, and maximal sentence length for all comparing methods. We also fix the GPT2 model (using GPT2-small) and the optimization algorithm (Adam). We only search the hyper-parameters of the initial learning rate of the Adam, where we conduct a systematic search for all comparing methods (within a range of 0.01∼0.00001).
>
>   For the hyper-parameters in our SeqPATE (irrelative to other baseline methods), we conducted the analyses on the teacher number and the number k in top-k selection and we reported them in the paper. When we were implementing our proposed strategies, we tuned the strategy-specific parameters. For the factor $\lambda$ balancing the KL loss and NLL loss, we heuristically tried {0.1, 1, 10, 20, 50, 100}. For the threshold $\mu$ in the top-$p$ strategy, we tried {0.90, 0.95, 0.98}. After we determine to use a specific strategy, we will not tune the hyper-parameter. We have clarified it in the current version.
>
>
> Q4.Typo: Should the grey line in Figure 2 be Pub-GPT + $\tilde{D}^{pub}$?
>
> Yes, it should be Pub-GPT + $\tilde{D}^{pub}$. We have fixed it in the current version. Thanks!

---

> > ### Comment · Reviewer_386M · 2021-11-22
> > **Hyperparameters for DPSGD**
> >
> > Thanks for the clarifications! However, I am still not convinced by the authors reply that DPSGD performs worse than their method SeqPATE at the same privacy levels. In Q2a in the common response to all reviewers, the authors show the performance of DPSGD with batch sizes of 32 and 128 and claim that SeqPATE beats DPSGD. But as Li et al. (2021) show, we need large batch sizes (such as 2048) for DPSGD to be competitive and requires tuning the learning rate carefully.
> > *When comparing two methods, you cannot compare them with the same hyperparameters. You need to choose the best parameters for each of them individually.*
> > The ghostclipping method from Li et al (2021) is only needed to speedup the implementation of DPSGD. If you have enough time, you can also implement DPSGD using batch size 2048 by gradient accumulation (i.e., by maintaining a partial sum of clipped gradients, you can implement any batch size). Saying that your GPU can only handle a batch of size 128 is not a valid reason.
> >
> > If the authors can compare and show that they beat a DPSGD benchmark from Li et al (2021) (where they already chose the best hyperparameters for DPSGD), using the best hyperparameters for SeqPATE on the same task and at the same privacy levels, I will be convinced that SeqPATE beats DPSGD. And I will be happy to revise my score.

---

> > > ### Author Response · Authors · 2021-11-23
> > > **Additional Experiments about [Li et al. 2021]**
> > >
> > > Thank you very much for your patience and suggestions.
> > >
> > > Even if the time is very tight, we will try our best to compare the concurrent work [Li et al. 2021] with our SeqPATE in a same setting until Nov. 29 (the deadline of reviewers' discussion).

---

> > > > ### Comment · Reviewer_386M · 2021-11-29
> > > > **Question on Additional Experiments about [Li et al. 2021]**
> > > >
> > > > Thanks for doing the additional experiments I requested. I have a few questions about the comparison between  DP-SGD + GhostClipping and SeqPATE in your table in point (1). I would like to know additional details about this experiment. In your comparison, did you train the DPSGD model on D^{priv} from (a) scratch (b) pre-trained GPT2 model or (c) pre-trained GPT2model + fine-tuned on D_{pub}?
> > > >
> > > > I think (c) should do the best, because it is using the information from D_{pub} and pre-trained GPT2, as is done by SeqPATE. So (c) should be a fair comparison.

---

> > > > > ### Author Response · Authors · 2021-11-30
> > > > > **Details about the New Experiments**
> > > > >
> > > > > Yes, we did (c) in our new experiment, where we fine-tune on the $\tilde{\mathcal{D}}^{pub}$ from the pre-trained GPT2 model, and then we continue to train DP-SGD on the private set $\mathcal{D}^{pri}$. The usage is the same as the row 5 (Pub-GPT+DP-SGD+$\tilde{\mathcal{D}}^{pub}$) in Table1 of our paper.
> > > > >
> > > > > Thank you very much.

---

> > > > > > ### Author Response · Authors · 2021-11-30
> > > > > > **Thanks for your comments!**
> > > > > >
> > > > > > Thank you very much for your comments and time!
> > > > > >
> > > > > > May I know are you satisfied with our explanation and new experiments. Is there anything else we need to address? We will keep the discussion forum open throughout the meta-review period in order to accommodate any last-minute discussions for such cases. If the discussion forum is not available for the authors, we will respond on our anonymous Github (github.com/anonymityForPaperSubmit123/ICLR22SeqPATE).

---

> > > > > > > ### Comment · Reviewer_386M · 2021-12-06
> > > > > > > **Updated my score to "marginally above acceptance threshold"**
> > > > > > >
> > > > > > > I am reasonably satisfied (though not fully) with the new experiments reported by the authors. So I am updating my score slightly as promised.

---

> > > > > > > > ### Author Response · Authors · 2021-12-06
> > > > > > > > **Thank you!**
> > > > > > > >
> > > > > > > > Thank you very much for your suggestions and time. We will add the new experiments to the next version and solve the issues you were concerned about.

---

> > > ### Author Response · Authors · 2021-11-29
> > > **Additional Experimental Results about [Li et al. 2021]**
> > >
> > > 1. We compare SeqPATE with **DP-SGD (using ghost clipping and batch size of 2048) [Li et al., 2021]** on the AirDialog dataset, and the experimental results show that **SeqPATE still outperforms DP-SGD** with $\varepsilon=3, \delta=1e-6$. We will add this result in the next version. Our implementation is based on a public GitHub project of ghost clipping [github.com/lxuechen/private-transformers], where we tune the learning rate from 0.01~0.00001, which is a symmetric hyper-parameter search compared to SeqPATE. In this experiment, we additionally tune the inference hyper-parameters of DP-SGD, since the BLEU score of the DP-SGD is quite low without tuning inference hyper-parameters. We have tried beam-search size (1,5,10), top-k (k=1, 20, 40), top-p (p=0.8, 0.9, 1).
> > >
> > >     |                     |                   | PPL   | BLEU-1 | BLEU-2 | BLEU-3 | BLEU-4 |
> > > | ------------------- | ----------------- | ----- | ------ | ------ | ------ | ------ |
> > > | DPSGD+ghost clipping | batchsize=2048,  $\varepsilon$=3, $\delta$=1e-6 | 11.17 | 13.21  | 5.90   | 3.15   | 1.54   |
> > > | SeqPATE(ours) | batchsize=128,  $\varepsilon$=3, $\delta$=1e-6 | 9.70  | 15.71  | 7.94   | 4.47   | 2.61   |
> > >
> > >
> > > 2. We're working on the E2E datasets (data-to-text) used in [Li et al., 2021]. Since the time is limited, we **will update results to our anonymous GitHub** [github.com/anonymityForPaperSubmit123/ICLR22SeqPATE], in case we finish it after the deadline of the authors’ responses (Nov. 29).
> > >
> > >     We note that **the data-to-text task is different from our setting and not very suitable for SeqPATE**. Our setting assumes that we have access to the sentence prefix as the public samples. We then use the public GPT2 model to generate pseudo sentences given the prefix. In data-to-text, the "prefix" corresponds to the "data" (i.e. table), thus without in-domain fine-tuning, we will not obtain reasonable pseudo data by autocompleting the prefix using the pre-trained GPT2 model.
> > >
> > >     Currently, SeqPATE works well on unconditional generation tasks (i.e. language modeling or sentence completion), thus is appropriate for language model-based pre-training on larger text. In future work, we will try to explore the usage of SeqPATE on other applications.
> > >
> > >
> > > 3. As for the hyper-parameter search, [Li et al., 2021] claimed that **they did not conduct too much hyper-parameter search** (only on the epoch number and batch size). The reason is that the hyper-parameter search of DP-SGD is based on the performance on the private validation set, which causes additional privacy loss but that loss was not counted in the reported results. [Li et al., 2021] also mentioned that point in their paper (Appendix B & I).
> > >
> > >     Similarly, in the experiments shown in our paper, we only search the learning rate on both SeqPATE and DP-SGD. Note that searching SeqPATE’s hyper-parameter is conducted on the public (student) validation set.
> > >
> > >
> > > 4. The fact that large-batch makes DP-SGD better is a **new finding** (even though it's not a new algorithm) that's **concurrent with our paper**. Even if the time is tight for new experiments, we tried our best to accomplish it and hope the results make our paper more convincing. Thank you very much for your suggestions.
> > >
> > >
> > > 5. Besides the above points, we also want to emphasize that SeqPATE’s advantages compared to DPSGD:
> > >
> > >     a) SeqPATE is a **model-agnostic framework**, thus can be easily adapted to non-gradient-based NLP methods (e.g. retrieval-based systems and in-context learning), where SGD is not applicable.
> > >
> > >     b) SeqPATE achieves **user-level privacy on secret phrases** without additional privacy cost by training each teacher on data from a single user, whereas DPSGD incurs a privacy cost that grows linearly with the number of occurrences of the secret phrase (as mentioned in Sec. 5.3). According to reviewer CVpe's suggestion, we conducted additional experiments to show SeqPATE's advantage in protecting users’ secret phrases. **The results show that SeqPATE can provide much stronger protection than DP-SGD (in terms of $\varepsilon$ for users’ secret phrases**, even if SeqPATE achieves a better utility. In the experiments, we treat “full name of the user” as secret phrases to protect. And different from the current SeqPATE, teachers’ training data are partitioned by users (each user’s samples occur in only one teacher).
> > >
> > >     |                     | $\varepsilon$ at sample level | $\varepsilon$ for users' secret phrases | PPL   | BLEU-1 | BLEU-2 | BLEU-3 | BLEU-4 |
> > > | ------------------- | ------------------------------  |-----------------------------------------------  | ----- | ------ | ------ | ------ | ------ |
> > > | DPSGD | 3 | $\textbf{123}$ |13.87 | 10.24  | 4.95   | 2.83   | 1.74 |
> > > | SeqPATE(ours) | 3 | $\textbf{3}$ | 11.45 | 12.83 | 6.19   | 3.33   | 1.86   |
> > >
> > >     c) SeqPATE achieves higher utility with strong privacy protection ($\varepsilon \leq 2$).
> > >
> > >
> > > [Li et al., 2021] Large Language Models Can Be Strong Differentially Private Learners

---

### Author Response · Authors · 2021-11-20
**Common responses to all reviewers [Part1]**

To **reviewer 1,2,3 (reviewer 386M, wFLx, and CVpe)** [Part1]:

**Q1**: Our utility is far from the non-private utility, which is surprising given results in two recent papers (ICLR submissions) on DPSGD for NLG. [Part1]

Thanks for pointing out the two recent papers [Yu et al., 2021a, Li et al., 2021] submitted to ICLR22. We have added discussion on these two papers in related work. We clarify the reasons for the seemingly discrepant results below.

1. **Difference in privacy levels.** We adopted stronger privacy protection at  $\varepsilon$=2 and $\delta$=1e-9, whereas Li et al. (2021) use $\varepsilon$=3 and $\delta$=1e-6, and Yu et al. (2021a) use $\varepsilon$=5.4 and $\delta$=1e-5. As a result, the utility gap between their private training and non-private training is smaller than ours. We have added new experiments using $\varepsilon$=3 and $\delta$=1e-6, and obtained similar results.
|                     |                   | PPL   | BLEU-1 | BLEU-2 | BLEU-3 | BLEU-4 |
| ------------------- | ----------------- | ----- | ------ | ------ | ------ | ------ |
| DPSGD | $\varepsilon$=2, $\delta$=1e-9 (original paper) | 17.65 | 7.90   | 2.69   | 1.14   |0.56 |
| DPSGD | $\varepsilon$=3, $\delta$=1e-6 (new setting)    | 13.87 | 10.24  | 4.95   | 2.83   | 1.74 |
| SeqPATE(our) | $\varepsilon$=2, $\delta$=1e-9 (original paper) | 13.67 | 11.56 | 4.35   | 1.82   | 0.78   |
| SeqPATE(our) | $\varepsilon$=3, $\delta$=1e-6 (new setting) | 9.94 | 15.55 | 7.82   | 4.37   | 2.54   |

    We would like to add that most DP researchers would agree that theoretically meaningful privacy parameters are those with $\varepsilon$ ≈ 1  and $\delta$ << 1/n. For $\varepsilon$ substantially larger than 1, the privacy guarantee deteriorates exponentially. Quoting co-inventor of DP Frank McSherry: "Anything much bigger than one is not a very reassuring guarantee," McSherry says. "Using an epsilon value of 14 per day strikes me as relatively pointless" as a safeguard.  See https://www.wired.com/story/apple-differential-privacy-shortcomings/
For these reasons, we find it compelling to report results for parameter regimes that DP researchers consider reasonable, despite the larger drop in the utility.  It is a great point that you raised that we should also report results on the weaker privacy regimes to be comparable to the concurrent literature. The above results have been added to the paper.

2. **Difference in DPSGD implementation.** The vanilla DPSGD without their improvements (which we used as baselines in this submission) is also unsatisfactory as reported in [Li et al., 2021; Yu et al., 2021a].

    a) Li et al. (2021) propose ghosting clipping to reduce memory usage so that the model can be trained with larger batch size. In the E2E (table-to-text generation) dataset, the vanilla DPSGD with a small batch size (32) only achieves 48.51 in BLEU, and only when using a large batch size (2048) does it reach 61.17, while non-private training gets 69.46. In our setting, the maximum batch size we can set is 128 under V100 (32G GPU memory). We were using batch size 32 in our experiments but if we increase the batch size to 128, we can also observe the performance is further improved as follows based on the results mentioned in “1. Difference in privacy levels”.

    |                     |                   | PPL   | BLEU-1 | BLEU-2 | BLEU-3 | BLEU-4 |
    | ------------------- | ----------------- | ----- | ------ | ------ | ------ | ------ |
    | DPSGD | batchsize=32,  $\varepsilon$=3, $\delta$=1e-6 | 13.87 | 10.24  | 4.95   | 2.83   | 1.74   |
    | DPSGD | batchsize=128,  $\varepsilon$=3, $\delta$=1e-6 | 12.75 | 10.87  | 5.41   | 3.02   | 1.98   |
    | SeqPATE(ours) | batchsize=32,  $\varepsilon$=3, $\delta$=1e-6 | 9.94  | 15.02  | 7.47   | 4.14   | 2.38   |
    | SeqPATE(ours) | batchsize=128,  $\varepsilon$=3, $\delta$=1e-6 | 9.70  | 15.71  | 7.94   | 4.47   | 2.61   |

    b) Yu et al. (2021a) apply new fine-tuning techniques (e.g. LoRA) to DPSGD and obtain good performance on the E2E dataset. However, we see that vanilla DPSGD results in poor performances on other datasets. For example, on MNLI, vanilla DPSGD gets 53.1 accuracy, Yu et al., (2021a)'s methods range from 80 to 83, and non-private training gets 87.6. On QQP, vanilla DPSGD gets 74.4, Yu et al. (2021a)'s methods range from 84 to 85, and non-private training gets 91.9. In summary, our experimental results are consistent with their results with respect to vanilla DPSGD.

---

> ### Author Response · Authors · 2021-11-20
> **Common responses to all reviewers [Part2]**
>
> To **reviewer 1,2,3 (reviewer 386M, wFLx, and CVpe)** [Part2]:
>
> **Q1**: Our utility is far from the non-private utility, which is surprising given results in two recent papers (ICLR submissions) on DPSGD for NLG. [Part2]
>
> 3. **Clarification on the gap between our utility and the non-private utility.** On Europarlv6, our utility gap is not too large (in table1 of our paper): In terms of B-1, non-private training gets 12.97; SeqPATE and DPSGD get 13.64 and 12.37 respectively, which aren't too far from the non-private utility. In comparison, [Li et al., 2021] also has a relatively large utility gap on DART (from 42.7 to 31.0). The utility gap on AirDialog seems large. The second reason is the non-private model on AirDialog has a strong ability to memorize the training data, that is the private leakage is very serious. So, to achieve the same level of protection (same $\varepsilon$), we should “pay out” more utility loss to avoid memorizing the training data. In table2, P-3 and P-4 achieve very high (0.94 and 0.91), which indicates most of the generated 3-grams and 4-grams come from the training set. It shows private leakage is very serious on the non-private model on AirDialog.
>
> 4. **Why we wanted to look beyond DPSGD.** The two concurrent works were released on Arxiv after the ICLR submission deadline so we weren’t able to compare against them. While the two papers have done a great job applying DPSGD to natural language generation (NLG) and natural language understanding (NLU), we believe our method adds new knowledge to the field too. As reviewer wFLx pointed out, the area of DP for NLG is underexplored and we need to push in different directions. Our algorithm is based on a **model-agnostic** framework, which works in settings where DPSGD is not applicable. For example, many recent NLP models rely on retrieval or in-context learning where no gradient update is used. As mentioned in Sec. 5.3, SeqPATE can **protect the secret phrases at the user level with few additional privacy losses**, but DP-SGD does not have such merit. Additionally, in our experiments, the PATE-based model achieves higher utility with small $\varepsilon$ (strong privacy protection).
>
> To **reviewer 2,3 (reviewer wFLx and CVpe)**:
>
> **Q2**: The computational cost of SeqPATE is high.
> Our implementation of SeqPATE runs on V100 (32GB GPU memory). Below we describe the training time and memory usage of SeqPATE.
>
> 1. **Memory usage and hard-disk space usage.**
> Since our method uses a large number of teachers, the naive implementation of loading all teachers into memory for aggregation is impractical. However, note that our algorithm only needs to access each teacher’s top-k prediction.
> Therefore, we train teacher models sequentially. Once a teacher model is trained, we obtain its top-k prediction (k=200 in our experiments) on the public training data and save the results (i.e. k probabilities). Then, we discard the teacher model. Finally, SeqPATE only needs teacher supervision on a small number of samples. In our experiments, training on 1-2k teacher labeled samples is sufficient. Overall, saving teachers’ inference results uses 8-16G. The memory usage is similar to that of a GPT2 model, because we do not load all teacher models into memory and instead run inference sequentially and merge teachers’ predictions offline.
>
> 2. **Training time.**
> While we have a large number of teachers, each teacher is trained on only a small fraction of the entire dataset. Thus the time it takes to train all teachers is roughly equal to the time of training a single GPT2 model on the full dataset (of 1~2M samples).
>
> In summary, with simple tricks, **the teacher training and aggregation steps are not much more expensive than training a GPT2 model.** Compared to standard NLG model training, our algorithm does not require special hardware or distributed learning. We have added the above discussion to the Appendix of our paper.

---

> > ### Author Response · Authors · 2021-11-20
> > **Common responses to all reviewers [Part3]**
> >
> > To **reviewer 2,4 (reviewer wFLx, CVpe, and pnSu)**:
> >
> > **Q3**: The definition and the motivation of the metrics P-3 and P-4.
> >
> > 1. **P-N is the fraction of unique n-grams in model generations that can be found in the private text.** The definition and our implementation are described in Appendix E and Eq. 4. We added the definition to the main paper in the current version.
> >
> > 2. **Reason for using P-3 and P-4**: One form of privacy leakage in NLG is to directly generate the original text spans in the training data, e.g., sensitive information such as addresses and SSNs may be leaked through the generated outputs. Avoiding leakage of private information in the output is also one of the main goals in recent work on private DP learning models in NLG [Kerrigan et al., 2020, Li et al., 2021; Yu et al., 2021a], including our paper. Therefore, we use P-n to calculate the fraction of generated n-grams that are also contained in the training corpus.
> >
> > 3. **We also conduct experiments on another privacy metric that uses a threat model [Carlini et al., 2019]**. The results are consistent with the results on P-3 and P-4. For example, in both metrics, the privacy leak in AirDialog is more serious than that in Europarl_v6 and the gap between Pri-GPT and SeqPATE on AirDialog is larger than that on Europarl_v6. (Please refer to the Appendix of our paper for more detailed experiment setups.)
> >
> >     |                   | AirDialog  | Europarl_v6 |
> >     | ----------------- | ----- | ------ |
> >     | Pri-GPT      | 3.13      | 2.83        |
> > | SeqPATE(ours) | 0.91      | 1.55        |
> >
> > To **reviewer 2,3,4 (reviewer CVpe and pnSu)**:
> >
> > **Q4**: The advantage of our model vanished given a large $\varepsilon$ compared to DPSGD.
> >
> > 1. **$\varepsilon$ = 0.1~5 is a reasonable and meaningful range of epsilon for privacy guarantee [Kamath et al., 2021]**. $\varepsilon$ > 5 says that an individual could be identified with a confidence more than 99.33% [Triastcyn et al., 2020].  We highlight that SeqPATE outperforms DPSGD in the region of strong privacy protection ($\varepsilon$ < 5), which provides meaningful protection in real-world applications. While most papers report results with $\varepsilon$ = 5, this privacy level is not acceptable in practice. The utility of DPSGD increases as $\varepsilon$ approaches infinity because the noise approaches 0. On the other hand, SeqPATE still suffers from utility loss due to distillation.
> > 2. However, **we note that this can be improved by using fewer teachers**. We conduct experiments with fewer (i.e. 100)  teachers with large $\varepsilon$ (i.e. 50 and 500) as follows, where SeqPATE outperforms DPSGD. We've added the experiments to the Appendix of our paper.
> >
> >     |                     |                   | PPL   | BLEU-1 | BLEU-2 | BLEU-3 | BLEU-4 |
> >     | ------------------- | ----------------- | ----- | ------ | ------ | ------ | ------ |
> >     | DPSGD | $\varepsilon$=50  | 11.21 | 9.68   | 4.45   | 2.43   | 1.43   |
> >     | SeqPATE (#teacher=100\) | $\varepsilon$=50 | 10.03 | 10.50  | 4.66   | 2.54   | 1.47   |
> >     | DPSGD | $\varepsilon$=500    | 10.86 | 9.81   | 4.52   | 2.49   | 1.45   |
> >     | SeqPATE (#teacher=100\) | $\varepsilon$=500 | 9.86  | 15.82  | 8.15   | 4.64   | 2.75   |
> >
> > [Li et al., 2021] Large Language Models Can Be Strong Differentially Private Learners.
> >
> > [Yu et al., 2021a] Differentially Private Fine-tuning of Language Models.
> >
> > [Kamath et al., 2021] Algorithms for Private Data Analysis (Intro to Differential Privacy).
> >
> > [Kerrigan et al., 2020] Differentially Private Language Models Benefit from Public Pre-training.
> >
> > [Carlini et al., 2019] The Secret Sharer: Evaluating and Testing Unintended Memorization in Neural Networks.
> >
> > [Triastcyn et al., 2020] Bayesian Differential Privacy for Machine Learning.

---

### Decision · Program_Chairs · 2022-01-20

**Decision:**

Reject

**Comment:**

This was a very borderline decision. Here are the major factors involved in the decision.

1. The concurrent works by Li et al and Yu et al. It is unclear about the relationship/strength between those results and the ones in the present paper. However, in accordance with the ICLR policy on simultaneous work, we ignore them to the extent possible.
2. The novelty of the approach. All reviewers agreed that the modifications to the PATE approach are fairly minor or incremental, and this appears to be largely an application of this method. That said, methods for the natural language setting are important.
3. The strength of the findings. Reviewers were mixed on the strength of the results: they appeared significant in some cases but rather lackluster in others.
4. The poor quality of the writing. I personally read the original version and found the writing to make it impossible to understand in parts. The writing is still subpar in many places, which I would have expected the authors to fully polish given the substantial time during the response period.

While in isolation, any of 2, 3, and 4 may be acceptable, their combination makes it difficult to recommend this paper for acceptance at this time.